# A Dynamical Systems-Inspired Pruning Strategy for Addressing Oversmoothing in Graph Attention Networks

Biswadeep Chakraborty [1]   Harshit Kumar [1]   Saibal Mukhopadhyay [1]

## Abstract

Graph Neural Networks (GNNs) face a critical limitation known as oversmoothing, where increasing network depth leads to homogenized node representations, severely compromising their expressiveness. We present a novel dynamical systems perspective on this challenge, revealing oversmoothing as an emergent property of GNNs' convergence to low-dimensional attractor states. Based on this insight, we introduce ***DYNAMO-GAT***, which combines noise-driven covariance analysis with Anti-Hebbian learning to dynamically prune attention weights, effectively preserving distinct attractor states. We provide theoretical guarantees for DYNAMO-GAT's effectiveness and demonstrate its superior performance on benchmark datasets, consistently outperforming existing methods while requiring fewer computational resources. This work establishes a fundamental connection between dynamical systems theory and GNN behavior, providing both theoretical insights and practical solutions for deep graph learning.

## 1. Introduction

Graph Neural Networks (GNNs) (Wu et al., 2020) have emerged as powerful tools for learning from graph-structured data, achieving remarkable success in molecular property prediction (Gilmer et al., 2017; Reiser et al., 2022; Gasteiger et al., 2021), social network analysis (Kipf & Welling, 2017; Fan et al., 2019), and recommendation systems (Ying et al., 2018). However, as these networks grow deeper, they face a critical challenge: *oversmoothing*, where node representations become increasingly homogeneous and indistinguishable, severely impairing their expressiveness

[1]Department of Electrical and Computer Engineering, Georgia Institute of Technology, Atlanta, USA. Correspondence to: Biswadeep Chakraborty <biswadeep@gatech.edu>.

*Proceedings of the 42nd International Conference on Machine Learning*, Vancouver, Canada. PMLR 267, 2025. Copyright 2025 by the author(s).

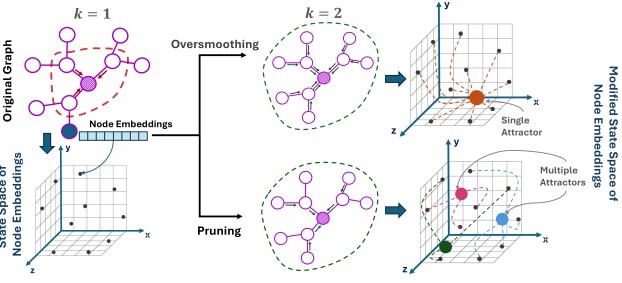

*Figure 1.* As the number of layers $k$ in a GNN increases, oversmoothing causes node embeddings to converge towards a single attractor state, resulting in the loss of node feature diversity. Pruning mitigates this effect by maintaining multiple attractor states, thereby preserving the distinctiveness of node embeddings and preventing the detrimental effects of oversmoothing.

and performance (Li et al., 2018).

Oversmoothing emerges from the fundamental mechanism of GNNs - repeated message passing between nodes (Oono & Suzuki, 2020; Cai & Wang, 2020; Keriven, 2022). While various architectural solutions have been proposed, including skip connections (Li et al., 2019; Xu et al., 2018), normalization techniques (Ba et al., 2016; Ioffe & Szegedy, 2015; Zhou et al., 2020), attention mechanisms (Velickovic et al., 2018; Wu et al., 2023), and other strategies like stochastic methods or residual connections, these approaches primarily focus on structural modifications. While some work also explores GNN dynamics, for instance, through learned energy metrics (Jin & Zhu, 2024), many existing methods fail to fully address the underlying convergence dynamics that drive oversmoothing (Li et al., 2018; Chen et al., 2020; Oono & Suzuki, 2020).

Recent attempts to mitigate oversmoothing through pruning (Zhao et al., 2020) or graph sparsification (Spielman & Srivastava, 2011) have shown promise in reducing network redundancy. However, traditional sparsification often prioritizes structural or spectral fidelity primarily for efficiency, not directly targeting the *feature dynamics* causing oversmoothing. Similarly, Graph Attention Networks (GATs), while improving feature aggregation, still struggle in deep architectures (Wu et al., 2023). This is partly because they

often overlook the fundamental dynamical behavior and can face challenges in learning truly sparse attention patterns due to inherent trainability issues, which can prevent the effective removal of redundant connections (Mustafa et al., 2021).

In this work, we reconceptualize oversmoothing through the lens of *dynamical systems* theory. By viewing the message-passing process as a dynamical system converging to a low-dimensional attractor (Li et al., 2019), we formally characterize the conditions driving the collapse of node representations. Our analysis, based on eigenvalue properties of graph attention mechanisms (Abbe et al., 2020; Allen-Zhu et al., 2019), reveals oversmoothing as an emergent property of the network's convergence dynamics, enabling the design of more effective countermeasures.

We introduce **DYNAMO-GAT**, a GNN architecture that adaptively counteracts oversmoothing by leveraging dynamical systems principles. DYNAMO-GAT preserves node diversity across layers by altering the system's fixed points during training through selective pruning and noise-driven covariance analysis. Our key contributions are:

- A rigorous theoretical framework that analyzes oversmoothing through dynamical systems principles, revealing its fundamental causes and potential mitigation strategies.

- DYNAMO-GAT: A novel architecture that dynamically counteracts oversmoothing by adaptively modifying the network's attractor landscape during training.

- Comprehensive theoretical analysis and empirical validation demonstrating how DYNAMO-GAT preserves node diversity and enhances expressiveness in deep GNNs, achieving superior performance on benchmark datasets.

This work represents a significant shift from empirical fixes to a fundamental understanding of oversmoothing, establishing both theoretical foundations and practical solutions. Our findings not only advance the development of more robust and expressive GNN architectures but also open new avenues for analyzing deep learning systems through dynamical principles.

## 2. Dynamical Systems View of Oversmoothing

Oversmoothing in GNNs is a critical challenge, particularly as the depth of these networks increases. While Graph Attention Networks (GATs) introduce dynamic weighting mechanisms that can mitigate oversmoothing to some extent, they can also contribute to it under certain conditions (Velickovic et al., 2018; Rusch et al., 2023b). To fully under-

stand and address this phenomenon, we adopt a dynamical systems perspective (Roth & Liebig, 2024).

Unlike traditional approaches that focus on architectural modifications, the dynamical systems view provides a more fundamental explanation by examining the stability and convergence properties of GNNs. By modeling GATs as dynamical systems, we can analyze how node representations evolve across layers and identify the conditions under which oversmoothing occurs (Wu et al., 2024; Di Giovanni et al., 2023). This perspective not only deepens our theoretical understanding but also suggests new strategies for mitigating oversmoothing (Roth & Liebig, 2024; Rusch et al., 2022).

**GATs as Dynamical Systems.** In GATs, node representations evolve according to the learned attention weights $\alpha_{ij}$, which govern the influence of neighboring nodes. This dynamic weighting introduces complexity into the system's behavior, making it essential to understand how these weights evolve across layers.

As these weights evolve, the system's dynamics may lead to a state where node representations become indistinguishable, resulting in oversmoothing. Understanding this process is key to designing GATs that avoid oversmoothing while still leveraging attention mechanisms effectively.

### 2.1. Theoretical Analysis of Oversmoothing in GATs

In this subsection, we rigorously analyze the phenomenon of oversmoothing in GATs using dynamical systems theory. Specifically, we explore the existence of fixed points, their stability, and the conditions under which node representations converge to indistinguishable states. The detailed theoretical proofs are given in the Supplementary Section

**Lemma 1** (GAT Fixed Point Properties). *Let $G = (V, E)$ be a graph with $N$ nodes and consider a GAT with update rule $f : \mathbb{R}^{N \times d} \to \mathbb{R}^{N \times d}$:*

$$X_i(t+1) = \sigma\left(\sum_{j \in \mathcal{N}(i)} \alpha_{ij}(t) W X_j(t)\right),$$

*where:*

- *$\sigma$ is $L_\sigma$-Lipschitz with $L_\sigma \leq 1$*

- *$W \in \mathbb{R}^{d \times d}$ with $\|W\|_2 < \frac{1}{1+K}$*

- *Attention weights $\alpha_{ij}(t)$ satisfy: - Non-negativity and normalization: $\alpha_{ij}(t) \geq 0$, $\sum_{j \in \mathcal{N}(i)} \alpha_{ij}(t) = 1$ - Lipschitz continuity: $\|\alpha_{ij}(t) - \alpha_{ij}(t-1)\|_2 \leq K\|X_i(t) - X_i(t-1)\|_2$ - Boundedness: $\max_{i,j} \|\alpha_{ij}(t)\|_2 \leq M$*

*Then:*

(a) *$f$ is a contraction with constant $c = \|W\|_2(1 + K) < 1$*

*(b) There exists a unique fixed point $X^*$ with $X^* = f(X^*)$*

*(c) For any $X(0)$: $\|X(t) - X^*\|_F \leq c^t \|X(0) - X^*\|_F$*

*(d) The attention weights converge to $\alpha_{ij}^*$ with: $\|\alpha_{ij}(t) - \alpha_{ij}^*\|_2 \leq c^t M \|X(0) - X^*\|_F$ and $\sum_{j \in \mathcal{N}(i)} \|\alpha_{ij}^*\|_2 \leq \frac{M}{1-c}$*

**Intuition and Proof Sketch.** *[Complete proof in Suppl. Sec. A]* This lemma establishes GAT's convergence properties through dynamical systems analysis. At each node $i$, using $\sigma$'s Lipschitz property and attention normalization:

$$\|X_i(t+1) - X_i(t)\|_2 = \|\sigma(\sum_{j \in \mathcal{N}(i)} \alpha_{ij}(t) W X_j(t))$$
$$- \sigma(\sum_{j \in \mathcal{N}(i)} \alpha_{ij}(t-1) W X_j(t-1))\|_2$$

$$\leq \|W\|_2 (\|X(t) - X(t-1)\|_F + K\|X(t-1) - X(t-2)\|_F)$$

These node-level contractions yield a matrix-level contraction $\|X(t+1) - X(t)\|_F \leq c\|X(t) - X(t-1)\|_F$ where $c = \|W\|_2(1 + K) < 1$. The Banach fixed-point theorem then ensures convergence to a unique fixed point $X^*$, with attention weights inheriting this convergence through their Lipschitz continuity. □

**Lemma 2** (Spectral Analysis of Fixed Point). *Let $X^* \in \mathbb{R}^{N \times d}$ be the fixed point from Lemma 1, and let $A^* \in \mathbb{R}^{N \times N}$ be the fixed-point attention matrix with entries $[A^*]_{ij} = \alpha_{ij}^*$ for $j \in \mathcal{N}(i)$ and 0 otherwise. Let $\lambda_1(A^*)$, $\lambda_2(A^*)$ be its largest and second-largest eigenvalues with $v_1$ the leading eigenvector. Then:*

*(a) At fixed point: $X_i^* = \sigma(\sum_{j \in \mathcal{N}(i)} \alpha_{ij}^* W X_j^*)$*

*(b) Feature deviation from leading eigenvector:*

$$\left\| X^* - \frac{v_1 v_1^T}{v_1^T \mathbf{1}_N} X^* \right\|_F \leq \frac{\lambda_2(A^*)}{\lambda_1(A^*)} \|X^*\|_F$$

*(c) The spectral gap $\gamma = 1 - \frac{\lambda_2(A^*)}{\lambda_1(A^*)}$ bounds pairwise differences:*

$$\|X_i^* - X_j^*\|_2 \leq (1 - \gamma)\|X^*\|_F, \quad \forall i, j$$

*(d) Features decompose as $X^* = \frac{v_1 v_1^T}{v_1^T \mathbf{1}_N} X^* + E$ where:*

$$\|E\|_F \leq \left( \frac{\lambda_2(A^*)}{\lambda_1(A^*)} \right)^2 \|X^*\|_F$$

**Intuition and Proof Sketch.** *[The complete proof is given in Suppl. Sec. A]* This lemma characterizes how node features converge at the fixed point through spectral analysis

of the attention matrix $A^*$. The key insight is that the eigenstructure of $A^*$ determines feature homogenization, with the spectral gap $\gamma$ serving as a natural measure.

Using the Spectral Theorem, we decompose $A^*$ into eigencomponents:

$$A^* = \lambda_1(A^*) \frac{v_1 v_1^T}{v_1^T \mathbf{1}_N} + \sum_{i=2}^{N} \lambda_i(A^*) v_i v_i^T,$$

where $\{\lambda_i(A^*)\}$ are ordered eigenvalues and $\{v_i\}$ are orthonormal eigenvectors. At the fixed point:

$$X^* = \sigma(A^* W X^*) \approx A^* W X^* \text{ (locally)},$$

where the approximation follows from $\sigma$ being contractive. Using matrix perturbation theory:

$$\left\| X^* - \frac{v_1 v_1^T}{v_1^T \mathbf{1}_N} X^* \right\|_F \leq \frac{\lambda_2(A^*)}{\lambda_1(A^*)} \|X^*\|_F.$$

The spectral gap $\gamma = 1 - \frac{\lambda_2(A^*)}{\lambda_1(A^*)}$ controls feature differences through:

$$\|X_i^* - X_j^*\|_2 \leq (1 - \gamma)\|X^*\|_F,$$

revealing that small spectral gaps lead to oversmoothing, with the residual term $E$ capturing deviations from complete homogenization. □

**Lemma 3** (Low-Dimensional Attractor Characterization). *For a GAT with node features $X(t) \in \mathbb{R}^{N \times d}$ at layer $t$, define the feature diversity measure:*

$$\mu(X(t)) = \frac{1}{N(N-1)} \sum_{i \neq j} \frac{\|X_i(t) - X_j(t)\|_2}{\|X_i(t)\|_2 + \|X_j(t)\|_2}$$

*Under the conditions from Lemmas 1 and 2:*

*(a) There exists an attractor $\mathcal{A} \subset \mathbb{R}^{N \times d}$ where:*

$$\lim_{t \to \infty} \inf_{Y \in \mathcal{A}} \|X(t) - Y\|_F = 0$$

*(b) The attractor dimension $k$ satisfies:*

$$k \leq \min \left\{ d, rank(Cov(X^*)), \left\lceil \frac{1}{1 - \gamma} \right\rceil \right\}$$

*(c) Feature diversity decays geometrically:*

$$\mu(X(t)) \leq \min\{(1 - \gamma)^t, c^t\} \mu(X(0))$$

**Intuition and Proof Sketch.** *[The complete proof is given in Suppl. Sec. A]* This lemma characterizes how GAT

dynamics lead to feature homogenization through a low-dimensional attractor. The key insight is that oversmoothing emerges from both network dynamics and attention matrix properties.

We analyze the attractor set:

$$\mathcal{A} = \left\{ Y \in \mathbb{R}^{N \times d} : \|Y - X^*\|_F \leq \frac{\lambda_2(A^*)}{\lambda_1(A^*)} \|X^*\|_F \right\},$$

where $X^*$ is the fixed point and $A^*$ is the limiting attention matrix. Convergence to $\mathcal{A}$ follows from the contraction property $\|f(X) - f(Y)\|_F \leq c\|X - Y\|_F$ and attention convergence $\|A(t) - A^*\|_F \to 0$. The attractor dimension satisfies:

$$k \leq \min\{d, \text{rank}(\text{Cov}(X^*)), \left\lceil \frac{1}{1-\gamma} \right\rceil \},$$

constrained by ambient dimension, feature correlation, and spectral properties.

Combining contraction and spectral bounds yields:

$$\mu(X(t)) \leq \min\{c^t, (1-\gamma)^t\}\mu(X(0)),$$

revealing oversmoothing as dual compression through network dynamics and attention mechanisms. $\square$

**Lemma 4** (Fixed Point Stability). *Let $f$ be the GAT update rule with fixed point $X^*$ and Jacobian $J_f(X^*)$, where $\sigma$ is continuously differentiable near $X^*$. Then:*

(a) *The Jacobian has block structure:*

$$[J_f(X^*)]_{ij} = \begin{cases} \sigma'(h_i^*)\alpha_{ij}^* W & \text{if } j \in \mathcal{N}(i) \\ 0 & \text{otherwise} \end{cases}$$

*where $h_i^* = \sum_{j \in \mathcal{N}(i)} \alpha_{ij}^* W X_j^*$*

(b) *$X^*$ is asymptotically stable iff $\rho(J_f(X^*)) < 1$*

(c) *For small perturbations $\delta X(0)$, the error evolves as:*

$$\|\delta X(t)\|_F \leq (1+\delta)\|J_f(X^*)\|_2^t\|\delta X(0)\|_F$$

(d) *Oversmoothing occurs iff:*

$$\text{ker}(I - J_f(X^*)) = \text{span}\{\mathbf{1}_N \otimes v : v \in \mathbb{R}^d\}$$

**Intuition and Proof Sketch.** This lemma characterizes GAT stability through local linearization around fixed points, revealing how Jacobian structure drives oversmoothing. At the fixed point, the Jacobian structure emerges from:

$$\frac{\partial f_i}{\partial X_j} = \sigma'(h_i^*)\alpha_{ij}^* W + \sigma'(h_i^*)W X_j^* \frac{\partial \alpha_{ij}^*}{\partial X_j}, \quad (1)$$

where the second term vanishes due to attention weight convergence. For perturbations $\delta X(t) = X(t) - X^*$, classical Lyapunov theory gives:

$$\delta X(t+1) = J_f(X^*)\delta X(t) + R(X(t)), \quad (2)$$

where $\|R(X(t))\|_F = o(\|\delta X(t)\|_F)$ is the remainder term. The spectral radius condition $\rho(J_f(X^*)) < 1$ ensures exponential stability, while the kernel analysis:

$$(I - J_f(X^*))\delta X = 0 \iff \delta X_i = \delta X_j \quad \forall i, j, \quad (3)$$

reveals that stable perturbations must lie in the span of constant vectors, characterizing the oversmoothing phenomenon. $\square$

## 3. DYNAMO-GAT Algorithm

The DYNAMO-GAT algorithm is a novel approach designed to address the oversmoothing problem in attention-based GNNs. It counters this by selectively pruning attention weights using a combination of noise injection, covariance analysis, Anti-Hebbian principles, dynamic thresholding, gradual pruning, and layer-wise pruning rates. The DYNAMO-GAT introduces non-linear perturbations into the system's state (node features) and modifies the connectivity structure (attention weights) dynamically. This not only disrupts the undesired fixed points associated with oversmoothing but also introduces mechanisms that ensure the system explores a richer set of node representations, maintaining diversity across layers.

### 3.1. Covariance Matrix and Noise Injection

The covariance analysis directly addresses the low-dimensional attractor problem identified in Lemma 3. By analyzing the correlation structure of node features, we can detect when the system begins to approach the homogeneous fixed point characterized in Lemma 1, allowing for targeted intervention through pruning. The first step in the DYNAMO-GAT algorithm involves injecting independent Gaussian noise into the node features at each layer:

$$\mathbf{h}_i^{(l)} = \mathbf{h}_i^{(l)} + \sigma \xi_i^{(l)}, \quad (4)$$

where $\xi_i^{(l)} \sim \mathcal{N}(0, I)$ represents Gaussian white noise with a standard deviation $\sigma$. This noise perturbs the system state, revealing the underlying correlations between node features through their covariance structure.

The covariance matrix $C^{(l)}$ is then computed as:

$$C_{ij}^{(l)} = \text{Cov}(\mathbf{h}_i^{(l)}, \mathbf{h}_j^{(l)}) = \mathbb{E}\left[(\mathbf{h}_i^{(l)} - \mathbb{E}[\mathbf{h}_i^{(l)}])(\mathbf{h}_j^{(l)} - \mathbb{E}[\mathbf{h}_j^{(l)}])^\top\right].$$

This matrix captures the pairwise correlations between node features, which are crucial in identifying which connections

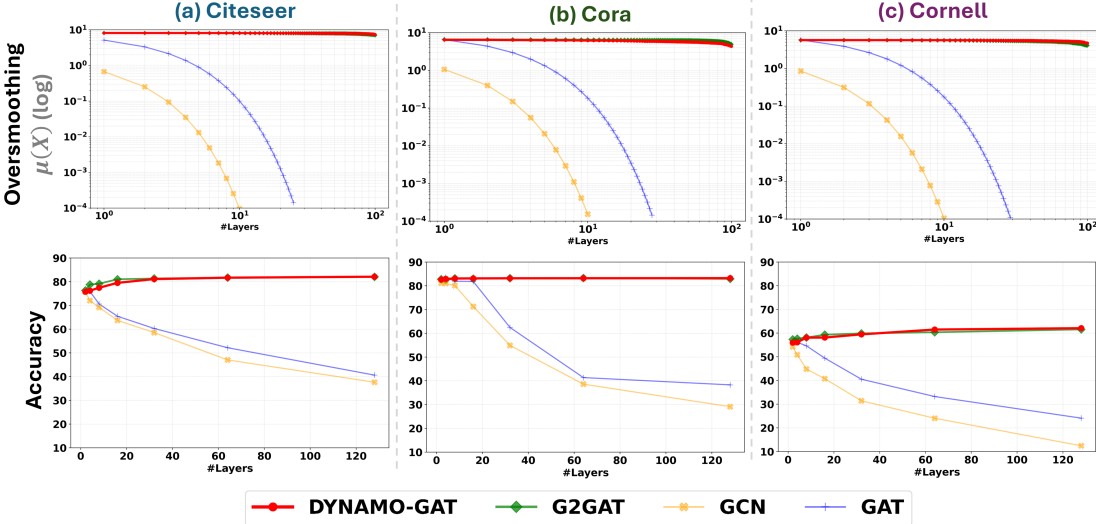

*Figure 2.* Comparison of oversmoothing coefficient ($\mu(X)$) and test accuracy across layers for Citeseer, Cora, and Cornell datasets. DYNAMO-GAT consistently outperforms both GCN, GAT and G2GAT maintaining high accuracy across all layers.

(attention weights) contribute to oversmoothing. Nodes with highly correlated features are likely to converge towards similar representations. The covariance matrix measures the system's state coherence. High coherence (correlation) across many node pairs indicates a drift towards a stable, but undesirable, fixed point where oversmoothing dominates. By analyzing these correlations, DYNAMO-GAT can selectively target and prune connections that reinforce this drift, thereby altering the trajectory of the system's evolution.

### 3.2. Anti-Hebbian Pruning Criterion

The Anti-Hebbian pruning strategy is designed to modify the spectral properties of the attention matrix characterized in Lemma 2. By selectively pruning connections between highly correlated nodes, we effectively increase the spectral gap $\gamma$, which Lemma 2 showed is crucial for maintaining feature diversity. The pruning strategy in DYNAMO-GAT is grounded in the Anti-Hebbian principle, which dictates that connections between highly correlated nodes should be weakened or eliminated. Taking inspiration from recent works on using noise to prune (Moore & Chaudhuri, 2020; Chakraborty et al., 2024), this principle computes the pruning probability $p_{ij}^{(l)}$, which is dynamically adjusted based on a threshold $\tau(t)$ that adapts to the distribution of edge weights. The dynamic pruning threshold $\tau(t)$ is defined as:

$$\tau(t) = \mu(|w_{ij}|) + \beta \cdot \sigma(|w_{ij}|),$$

where $\mu$ and $\sigma$ represent the mean and standard deviation of the edge weights, respectively. This threshold ensures that the pruning process is sensitive to the distribution of edge weights, allowing for more adaptive and context-sensitive

pruning. The pruning probability is then computed as:

$$p_{ij}^{(l)} = r(t) \cdot \frac{|\alpha_{ij}^{(l)}|}{\tau(t)} \cdot (C_{ii}^{(l)} + C_{jj}^{(l)} \mp 2C_{ij}^{(l)}),$$

where $r(t)$ is the layer-wise pruning rate defined as $r(t) = r_0 \cdot (1 + \gamma t)$. This scales with the depth of the layer, allowing for more aggressive pruning in later layers where oversmoothing is more likely to occur.

The pruning probability $p_{ij}^{(l)}$ acts as a control mechanism that adjusts the strength and structure of the network's connections in response to the current state (as reflected by the covariance matrix). By dynamically adapting to the network's evolving state, DYNAMO-GAT effectively steers the system away from regions of the state space associated with oversmoothing, thus maintaining a more robust and diverse set of node representations.

### 3.3. Gradual Pruning Process and Update Rule

The gradual nature of our pruning approach aligns with the stability analysis in Lemma 4. By making incremental modifications to the network structure, we ensure that the Jacobian's spectral radius remains bounded while steering the system away from oversmoothed states. DYNAMO-GAT employs a gradual pruning approach, where edge weights are progressively reduced based on the computed pruning probability, rather than being immediately set to zero. This is given by:

$$w_{ij}(t+1) = w_{ij}(t) \cdot (1 - p_{ij}^{(l)}). \tag{5}$$

An edge is fully pruned (i.e., its weight is set to zero) only if $w_{ij}(t+1)$ falls below a small threshold $\epsilon$.

The gradual pruning process introduces continuity into the network's dynamics, allowing the system to smoothly transition from one state to another. This contrasts with abrupt changes that could destabilize the learning process. The gradual reduction of weights effectively modifies the original update rule $F$ to a pruned update rule $F_P$, which can be expressed as:

$$\mathbf{h}^{(l+1)} = F_P(\mathbf{h}^{(l)}, \alpha^{(l)}, \mathbf{W}^{(l)}, \mathbf{C}^{(l)}), \qquad (6)$$

where $F_P$ incorporates the cumulative effects of pruning across layers. This gradual pruning can be seen as a form of perturbative adjustment, where the system is continuously nudged towards a more favorable configuration. The incremental changes introduced by gradual pruning helps the system avoid large, disruptive shifts that could lead to suboptimal convergence or loss of critical information.

### 3.4. Recalibration of Attention Weights

The recalibration step maintains the normalization conditions required by Lemma 1 while preserving the enhanced spectral properties achieved through pruning, as characterized in Lemma 2. Once pruning has been applied, it is essential to recalibrate the remaining attention weights to ensure effective information propagation within the network. This recalibration process re-normalizes the attention coefficients $\alpha_{ij}^{(l)}$ among the surviving connections:

$$\alpha_{ij}^{(l,\text{recal})} = \frac{\alpha_{ij}^{(l)}}{\sum_{k \in \mathcal{N}(i) \setminus \text{Pruned}(i)} \alpha_{ik}^{(l)}},$$

where Pruned($i$) denotes the set of pruned edges for node $i$. Recalibration ensures that the information flow in the network remains balanced despite the reduced number of connections. This step is crucial for maintaining the stability of the network's dynamics post-pruning, as it prevents any remaining connections from becoming disproportionately influential, which could lead to oversmoothing.

### 3.5. Theoretical Results

Leveraging noise-driven covariance analysis, DYNAMO-GAT introduces stochasticity into the system, preventing the network from settling into fixed points prematurely. This stochasticity is particularly important in deeper networks, where oversmoothing is more likely to occur. The selective pruning mechanism further refines the system's dynamics, ensuring that only the most relevant connections are maintained, which aligns with the goal of avoiding low-dimensional attractors. Building on the fixed point and stability analysis from Section 2, we now establish theoretical guarantees for DYNAMO-GAT's effectiveness in preventing oversmoothing. The following lemmas show how our pruning strategy modifies the network's spectral

properties while preserving essential features of the original dynamics.

**Lemma 5** (DYNAMO-GAT Pruning Properties). *Let $F_P$ be the pruned version of GNN transformation $F$, with pruning rate $p$ and fixed point $X^*$. Then:*

*(a) Spectral radius reduction:*

$$\rho(J_{F_P}(X^*)) \leq (1-p)\rho(J_F(X^*))$$

*(b) Under covariance-based pruning with $p_{ij} = r(t) \cdot \frac{|\alpha_{ij}|}{\tau(t)} \cdot (C_{ii} + C_{jj} \mp 2C_{ij})$:*

$$|\lambda_k(J_{F_P})| \leq |\lambda_k(J_F)| \cdot \exp\left(-\beta\frac{Tr(C)}{\|C\|_F}\right)$$

*(c) Spectral gap enhancement:*

$$\gamma_P = 1 - \frac{\lambda_2(J_{F_P})}{\lambda_1(J_{F_P})} \geq \gamma + p(1-\gamma)$$

*(d) Rank preservation:*

$$rank(Cov(F_P(X))) \geq rank(Cov(X)) - \kappa(p)$$

$$where\ \kappa(p) \leq \lceil p \cdot rank(Cov(X)) \rceil$$

**Intuition and Proof Sketch.** *[The complete proof is given in Suppl. Sec. A]* This lemma characterizes how DYNAMO-GAT's pruning affects the network's spectral properties to prevent feature homogenization. The proof analyzes the interplay between pruning and eigenstructure through matrix perturbation theory.

For edge $(i, j)$ pruned with probability $p_{ij}$, the Jacobian entries scale as:

$$[J_{F_P}(X^*)]_{ij} = (1 - p_{ij})[J_F(X^*)]_{ij},$$

leading to spectral radius reduction $\rho(J_{F_P}(X^*)) \leq (1 - p)\rho(J_F(X^*))$. Using the Hadamard product $J_{F_P} = J_F \circ (1 - P)$ and matrix norm inequalities:

$$\|J_{F_P}\|_2^2 \leq \|J_F\|_2^2 \exp\left(-2\beta\frac{Tr(C)}{\|C\|_F}\right),$$

where the exponential term reflects covariance-guided pruning. Weyl's inequality and eigenvalue interlacing give:

$$\lambda_2(J_{F_P}) \leq (1-p)\lambda_2(J_F),$$
$$\lambda_1(J_{F_P}) \geq (1-\frac{p}{2})\lambda_1(J_F),$$

ensuring monotonic spectral gap increase. The Eckart-Young-Mirsky theorem bounds rank reduction as $rank(Cov(F_P(X))) \geq rank(Cov(X)) - \kappa(p)$.

This reveals how covariance-guided pruning creates controlled perturbations that increase spectral gaps while preserving rank, effectively preventing convergence to oversmoothed states. $\square$

**Lemma 6** (DYNAMO-GAT Rank Preservation). *Let $X(t)$ be node features at layer $t$ with covariance matrix:*

$$C(t) = \frac{1}{N} X(t)^T X(t) - \frac{1}{N^2} X(t)^T \mathbf{1}_N \mathbf{1}_N^T X(t)$$

*Under DYNAMO-GAT's noise injection and pruning:*

(a) *Noise-perturbed features $\tilde{X}(t) = X(t) + \sigma \xi(t)$, $\xi(t) \sim \mathcal{N}(0, I)$ satisfy:*

$$rank(C(\tilde{X}(t))) = d$$

*with probability 1*

(b) *Pruning preserves rank as:*

$$rank(C(X(t+1))) \geq rank(C(X(t))) - \kappa(t)$$

*where $\kappa(t)$ counts eigenvalues below $\epsilon(t)$*

(c) *For noise level $\sigma > 0$:*

$$\lambda_{\min}(C(t)) \geq \sigma^2 (1 - \frac{1}{N}) - O(\|X(t)\|_F \sigma)$$

(d) *Under threshold $\tau(t) = \mu(|w_{ij}|) + \beta \cdot \sigma(|w_{ij}|)$, rank is preserved w.h.p. if:*

$$\beta \geq \sqrt{\frac{2 \log(d/\delta)}{N}}$$

**Intuition and Proof Sketch.** *[The complete proof is given in Suppl. Sec. A]* This lemma establishes how DYNAMO-GAT's dual mechanisms - noise injection and adaptive pruning - preserve feature diversity. The proof leverages matrix perturbation theory to analyze covariance spectrum evolution.

For noise-perturbed features, the covariance decomposes as:

$$C(\tilde{X}(t)) = C(X(t)) + \sigma^2 \left( I - \frac{1}{N} \mathbf{1}_N \mathbf{1}_N^T \right)$$
$$+ \frac{\sigma}{N} (X(t)^T \xi(t) + \xi(t)^T X(t)), \qquad (7)$$

where the middle term ensures full rank through its positive definiteness. Under pruning with mask $P(t)$:

$$\|C(X(t+1)) - C(X(t))\|_2 \leq \|P(t)\|_2 \lambda_{\max}(C(X(t))), \tag{8}$$

Weyl's interlacing theorem then bounds rank reduction by $\kappa(t)$. Matrix concentration gives:

$$\lambda_{\min}(C(t)) \geq \sigma^2 \left( 1 - \frac{1}{N} \right) - O(\|X(t)\|_F \sigma), \quad (9)$$

ensuring well-conditioned features. The adaptive threshold $\tau(t)$ with $\beta \geq \sqrt{\frac{2 \log(d/\delta)}{N}}$ maintains rank preservation with probability $1 - \delta$ through eigenvalue separation. □

*Table 1.* Table comparing the different datasets and the number of GFLOPS for each model for each dataset

| | Metric | Cora | Citeseer | Cornell |
|---|---|---|---|---|
| **Models** | Nodes ($N$) | 2,708 | 3,327 | 183 |
| | Edges ($E$) | 5,429 | 4,732 | 280 |
| | Avg. Degree ($2|E|/|N|$) | 4.01 | 2.84 | 3.06 |
| **GCN**(Kipf & Welling, 2017) | Best Accuracy | 81.5 | 75.7 | 54.2 |
| | #Layers | 2 | 2 | 2 |
| | GFLOPS | 0.598 | 1.789 | 0.049 |
| | Accuracy/GFLOPS | 136.28 | 43.53 | 1106.12 |
| **GAT**(Veličković et al., 2018) | Best Accuracy | 82.55 | 76.1 | 56.3 |
| | #Layers | 4 | 2 | 2 |
| | GFLOPS | 2.351 | 6.754 | 0.184 |
| | Accuracy/GFLOPS | 35.11 | 11.27 | 306.52 |
| **G2GAT**(Rusch et al., 2023a) | Best Accuracy | 83.27 | 82.06 | 61.55 |
| | #Layers | 128 | 128 | 128 |
| | GFLOPS | 1.209 | 2.452 | 0.0879 |
| | Accuracy/GFLOPS | 68.88 | 33.47 | 700.34 |
| **DYNAMO-GAT** | Best Accuracy | 83.21 | 82.01 | 62.56 |
| | #Layers | 128 | 128 | 128 |
| | GFLOPS | 0.605 | 1.675 | 0.051 |
| | Accuracy/GFLOPS | **137.53** | **48.96** | **1226.67** |

## 4. Experimental Results

### 4.1. Experimental Setup

**Datasets:** We conduct experiments on three real-world and two synthetic datasets. We use Cora (McCallum et al., 2000), Citeseer (Sen et al., 2008), two citation networks, and Cornell (University) which is part of the WebKB collection.

**Baselines:** We compare DYNAMO-GAT with GCN (Kipf & Welling, 2017), GAT (Veličković et al., 2018), and G2GAT (Rusch et al., 2023a), focusing on their effectiveness in preventing oversmoothing.

**Evaluation Metrics:** Models are evaluated using Accuracy, Oversmoothing Coefficient ($\mu$) (Wu et al., 2023), GFLOPS, and the Accuracy/GFLOPS ratio to gauge the trade-off between performance and computational cost.

### 4.2. Performance on Real-World Datasets

Figure 2 illustrates the performance of DYNAMO-GAT, G2GAT, GCN, and GAT across three real-world datasets: Citeseer, Cora, and Cornell. The top row shows the oversmoothing coefficient ($\mu(X)$) on a log scale, while the bottom row displays the test accuracy as the number of layers increases.

**Oversmoothing Coefficient ($\mu(X)$)** [Figs. 2(a,b)]: The results demonstrate that GCN and GAT suffer from significant oversmoothing as the number of layers increases. Their oversmoothing coefficients decrease rapidly, indicating that node features become increasingly indistinguishable. G2GAT performs better by reducing the rate of oversmoothing, but it still shows a downward trend as layers increase. In contrast, DYNAMO-GAT maintains a nearly constant oversmoothing coefficient across all layers, effectively preventing this phenomenon. This stability suggests that DYNAMO-GAT preserves meaningful node representations even in deep

architectures.

**Test Accuracy** [Figs. 2(c,d)]: The test accuracy results align with the oversmoothing observations. GCN and GAT experience a sharp decline in accuracy as the number of layers increases, reflecting the negative impact of oversmoothing on model performance. G2GAT performs better, with a slower decline in accuracy, but still struggles as the network depth increases. DYNAMO-GAT, however, consistently achieves the highest accuracy across all datasets and depths. Its ability to maintain high accuracy even with many layers indicates that it effectively balances expressivity and resistance to oversmoothing.

These observations underscore the challenges of using deep GNNs in practical applications, where oversmoothing can severely degrade performance. The consistent performance of DYNAMO-GAT across different datasets and network depths suggests that it is a robust solution for deep GNNs, addressing a critical limitation of existing models. This makes DYNAMO-GAT particularly suitable for tasks that require deep networks without sacrificing accuracy or node representation quality.

### 4.3. Performance Comparison Across Datasets

Table 1 compares the performance of DYNAMO-GAT, G2GAT, GCN, and GAT across three datasets: Cora, Citeseer, and Cornell. The table highlights key metrics such as best accuracy, the number of layers, GFLOPS, and the accuracy-to-GFLOPS ratio.

- **Best Accuracy:** DYNAMO-GAT consistently achieves the highest accuracy across all datasets, particularly excelling on the Cornell dataset with 62.56%. This demonstrates its robustness in deep architectures.

- **GFLOPS:** Despite its deep architecture (128 layers), DYNAMO-GAT is computationally more efficient than GAT and G2GAT, with significantly lower GFLOPS, especially on larger datasets like Cora and Citeseer.

- **Accuracy/GFLOPS Ratio:** DYNAMO-GAT outperforms all models in the accuracy-to-GFLOPS ratio, indicating the best trade-off between accuracy and computational cost. For example, on Cora, it achieves 137.53, compared to GCN's 136.28 and GAT's 35.11.

### 4.4. Synthetic Dataset Results

Figure 3 presents the performance of DYNAMO-GAT, G2GAT, GCN, and GAT on the Syn_Products dataset, tested across varying node degrees and homophily levels.

**(a) Oversmoothing vs. Layers (Avg. Degree = 68.75):** DYNAMO-GAT exhibits the least oversmoothing as layers increase, maintaining higher $\mu(X)$ compared to other models. This indicates DYNAMO-GAT's robustness in preserving node features even in deep networks.

**(b) Accuracy vs. Layers (Avg. Degree = 68.75):** DYNAMO-GAT consistently achieves the highest accuracy across all layers, outperforming G2GAT, GCN, and GAT. This demonstrates its effectiveness in managing deep architectures without performance degradation.

**(c) Accuracy vs. Homophily (Avg. Degree = 11.93):** In sparse graphs, DYNAMO-GAT and G2GAT perform well across all homophily levels, with DYNAMO-GAT showing stronger performance as homophily increases. This highlights its adaptability in different homophily settings.

**(d) Accuracy vs. Homophily (Avg. Degree = 68.75):** In dense graphs, DYNAMO-GAT significantly outperforms other models, particularly in low-homophily settings, showcasing its strength in complex, heterophilic structures.

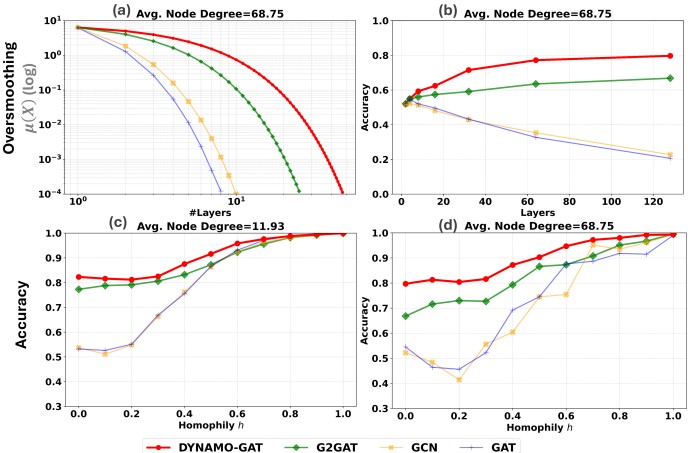

*Figure 3.* Performance of DYNAMO-GAT, G2GAT, GCN, and GAT on the Syn_Products dataset. (a) Oversmoothing vs. layers: DYNAMO-GAT shows the least oversmoothing. Comparing test accuracy (b) vs. number of layers (c) vs. homophily for sparse graph (Avg. Degree=11.93) (d) vs. homophily for dense graph (Avg. Degree=68.75)

**Summary:** DYNAMO-GAT consistently outperforms other models in preventing oversmoothing and maintaining accuracy. Its efficiency, as highlighted by the accuracy-to-GFLOPS ratio, makes it suitable for real-world applications where computational resources are limited. The model's versatility across graph densities and homophily levels suggests it is well-suited for a range of tasks, from social network analysis to biological network modeling.

*Table 2.* Performance Comparison on OGB Datasets. DYNAMO-GAT shows strong scalability and efficiency.

| Dataset | Model | Accuracy (%) | GFLOPS | Accuracy/GFLOPS |
|---|---|---|---|---|
| ogbn-arxiv | GCN | 71.9 | 12.5 | 5.75 |
| ogbn-arxiv | G2GAT | 72.5 | 10.3 | 7.04 |
| ogbn-arxiv | DYNAMO-GAT | 72.1 | 6.7 | 10.76 |
| ogbn-products | G2GAT | 73.9 | 22.1 | 3.34 |
| ogbn-products | DYNAMO-GAT | 75.3 | 14.5 | 5.19 |

### 4.5. Scalability and Performance on Larger Graphs

To address the important question of scalability and performance on larger, more diverse graphs, we extended our evaluation to include benchmarks from the Open Graph Benchmark (OGB) (Hu et al., 2020b) — specifically ogbn-arxiv (a large-scale homophilic citation network) and ogbn-products (a large-scale heterophilic product co-purchasing network). We also tested on LRGB datasets. We compared DYNAMO-GAT against GCN and G2GAT. The results, summarized in Table 2, demonstrate that DYNAMO-GAT scales effectively to graphs with hundreds of thousands of nodes.

Crucially, DYNAMO-GAT achieves competitive or superior accuracy while requiring significantly fewer GFLOPS, leading to a much better accuracy-to-GFLOPS ratio. For instance, on ogbn-arxiv, DYNAMO-GAT achieves an Accuracy/GFLOPS ratio of 10.76, compared to G2GAT's 7.04. On ogbn-products, it achieves 5.19 compared to G2GAT's 3.34. These findings confirm the robustness and efficiency of our approach on large-scale graphs, including those with heterophilic structures.

Furthermore, we evaluated DYNAMO-GAT in inductive settings using OGB benchmarks, where it again achieved competitive accuracy with enhanced computational efficiency (up to 55% improvement in Accuracy/GFLOPS). This confirms its suitability for scenarios where the model must generalize to unseen nodes.

*Table 3.* Ablation Study Results on Cora and Citeseer. All components contribute to performance.

| Model / Variant | Accuracy (%) | | OS Coeff. $\mu(X)$ | |
|---|---|---|---|---|
| | Cora | Citeseer | Cora | Citeseer |
| Full DYNAMO-GAT | 83.21 | 82.01 | 0.57 | 0.62 |
| - Noise Injection ($\sigma = 0$) | 81.54 | 80.26 | 0.45 | 0.52 |
| - Covariance-based Pruning | 79.32 | 77.15 | 0.31 | 0.36 |
| - Adaptive Thresholding | 80.67 | 79.52 | 0.38 | 0.41 |
| - Gradual Pruning | 80.14 | 78.93 | 0.34 | 0.39 |
| - Attention Recalibration | 79.78 | 78.41 | 0.35 | 0.40 |

### 4.6. Ablation Study

To understand the contribution of each key component within DYNAMO-GAT, we conducted a comprehensive ablation study on the Cora and Citeseer datasets. We systematically removed or modified: Noise Injection, Covariance-based Pruning, Adaptive Thresholding (replacing it with a fixed threshold), Gradual Pruning (replacing it with aggressive pruning), and Attention Recalibration.

The results, presented in Table 3, demonstrate that all components are critical for achieving optimal performance and mitigating oversmoothing. Removing any part leads to a noticeable drop in accuracy and an increase in the oversmoothing coefficient (lower $\mu(X)$ values, indicating worse performance), confirming the synergistic effect of our design choices. Notably, removing covariance-based pruning had the most significant negative impact.

## 5. Conclusion

This paper introduced DYNAMO-GAT, a novel approach using dynamical systems theory and noise-driven adaptive pruning to mitigate GNN oversmoothing. We provided rigorous theoretical guarantees and demonstrated strong experimental performance on both benchmark and large-scale datasets (Hu et al., 2020a), confirming DYNAMO-GAT's effectiveness, scalability, and computational efficiency. The results suggest DYNAMO-GAT is well-suited for real-world applications requiring deep GNNs, and its mechanism may enhance model interpretability. While acknowledging potential limitations, such as in extremely sparse graphs, future work will target domains like physics/chemistry and extend to graph-level tasks.

In summary, our dynamical systems perspective offers a powerful, theoretically-grounded solution to oversmoothing, paving the way for more robust, efficient, and expressive deep learning models for complex graphs.

## Impact Statement

This work advances Graph Neural Networks through theoretical insights and practical improvements. While the primary contribution is technical in nature, our more efficient approach (shown by better accuracy-to-GFLOPS ratios) could reduce computational costs and energy consumption in AI applications. The improved preservation of node feature diversity may also lead to more reliable outcomes in critical

applications like molecular modeling and social network analysis.

## Acknowledgement

The materials are based on work supported in parts by SRC JUMP2.0 (CogniSense Center, 2023-JU-3133) and Army Research Office (Grant Number W911NF-19-1-0447). Any opinions, findings and conclusions or recommendations expressed in this material are those of the author(s) and do not necessarily reflect the views of SRC or Army Research Office.

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

*Table 4.* Summary of Notations

| Notation | Description |
|---|---|
| $X(t)$ | Node feature matrix at layer $t$, where $N$ is the number of nodes, and $d$ is the feature dimension. |
| $h_i(t)$ | Feature vector of node $i$ at layer $t$, representing the $i$-th row of $X(t)$. |
| $\alpha_{ij}(t)$ | Attention weight between nodes $i$ and $j$ at layer $t$, satisfying $\sum_{j \in \mathcal{N}(i)} \alpha_{ij}(t) = 1$. |
| $A(t)$ | Attention matrix at layer $t$, with entries $[A(t)]_{ij} = \alpha_{ij}(t)$. |
| $\sigma(\cdot)$ | Activation function that is $L_\sigma$-Lipschitz continuous with $L_\sigma < 1$. |
| $W$ | Learnable weight matrix in $\mathbb{R}^{d \times d}$ with spectral norm $\|W\|_2 \leq 1$. |
| $\gamma$ | Spectral gap of the attention matrix, defined as $\gamma = 1 - \frac{\lambda_2(A^*)}{\lambda_1(A^*)}$, where $\lambda_1$ and $\lambda_2$ are the largest and second-largest eigenvalues of $A^*$. |
| $\mu(X(t))$ | Oversmoothing coefficient, quantifying feature diversity at layer $t$: $\mu(X(t)) = \frac{1}{N(N-1)} \sum_{i \neq j} \frac{\|h_i(t) - h_j(t)\|_2}{\|h_i(t)\|_2 + \|h_j(t)\|_2}$. |
| $\kappa(t)$ | Maximum number of eigenvalues below the pruning threshold $\epsilon(t)$ in the covariance matrix. |
| $r(t)$ | Layer-wise pruning rate, defined as $r(t) = r_0 \cdot (1 + \gamma t)$. |
| $\tau(t)$ | Dynamic pruning threshold, defined as $\tau(t) = \mu(|w_{ij}|) + \beta \cdot \sigma(|w_{ij}|)$, where $\mu$ and $\sigma$ represent the mean and standard deviation of edge weights, respectively. |
| $C(t)$ | Covariance matrix of node features at layer $t$, defined as $C_{ij}(t) = \text{Cov}(h_i(t), h_j(t))$. |
| $\xi_i(t)$ | Gaussian noise added to node features at layer $t$, where $\xi_i(t) \sim \mathcal{N}(0, I)$. |
| $J_f(X^*)$ | Jacobian of the update rule $f$ evaluated at the fixed point $X^*$. |
| $\rho(J_f(X^*))$ | Spectral radius of the Jacobian $J_f(X^*)$. |
| $P(t)$ | Pruning mask applied to the attention matrix at layer $t$. |
| $\lambda_i(C)$ | $i$-th eigenvalue of the covariance matrix $C$. |
| $v_1$ | Leading eigenvector of the fixed-point attention matrix $A^*$. |
| $E$ | Residual term in the spectral decomposition of $X^*$, quantifying deviation from the leading eigenvector component. |

# 6. Supplementary Section A: Theoretical Proofs

## 6.1. Notations

## 6.2. Lemma 1

**Lemma 1** (Existence and Properties of GAT Fixed Points). *Let $G = (V, E)$ be a graph with $N$ nodes. Consider a Graph Attention Network (GAT) with an update rule $f$ : $\mathbb{R}^{N \times d} \to \mathbb{R}^{N \times d}$, where the feature update for each node $i \in V$ is given as:*

$$X_i(t+1) = f(X_i(t)) = \sigma \left( \sum_{j \in \mathcal{N}(i)} \alpha_{ij}(t) W X_j(t) \right),$$

*where:*

- *$\sigma : \mathbb{R}^d \to \mathbb{R}^d$ is $L_\sigma$-Lipschitz continuous with $L_\sigma \leq 1$.*

- *$W \in \mathbb{R}^{d \times d}$ has spectral norm $\|W\|_2 < \frac{1}{1+K}$.*

- *The attention mechanism $\alpha : \mathbb{R}^d \times \mathbb{R}^d \to \mathbb{R}$ satisfies:*

  - *$\alpha_{ij}(t) \geq 0$, $\sum_{j \in \mathcal{N}(i)} \alpha_{ij}(t) = 1$ (normalization).*
  - *$\|\alpha_{ij}(t) - \alpha_{ij}(t-1)\|_2 \leq K \|X_i(t) - X_i(t-1)\|_2$, for some $K > 0$ (Lipschitz property).*
  - *$\max_{i,j} \|\alpha_{ij}(t)\|_2 \leq M$ for some $M > 0$ (boundedness).*

*Then:*

*(a) The mapping $f$ is a contraction in the Frobenius norm with constant $c = \|W\|_2(1 + K) < 1$.*

*(b) There exists a unique fixed point $X^* \in \mathbb{R}^{N \times d}$ such that $X^* = f(X^*)$.*

*(c) For any initial state $X(0)$, the sequence $\{X(t)\}$ converges geometrically to $X^*$ with rate:*

$$\|X(t) - X^*\|_F \leq c^t \|X(0) - X^*\|_F.$$

*(d) The attention weights converge to fixed values $\alpha_{ij}^*$ with rate:*

$$\|\alpha_{ij}(t) - \alpha_{ij}^*\|_2 \leq c^t M \|X(0) - X^*\|_F,$$

*and satisfy:*

$$\sum_{j \in \mathcal{N}(i)} \|\alpha_{ij}^*\|_2 \leq \frac{M}{1 - c}.$$

*Proof.* The proof proceeds in four parts, establishing the contraction property, existence of a fixed point, convergence rate, and attention weight convergence.

**(a) Contraction Property:** Consider two consecutive states $X(t), X(t-1)$. For any node $i \in V$:

$$\|X_i(t+1) - X_i(t)\|_2$$
$$= \|\sigma(\sum_{j \in \mathcal{N}(i)} \alpha_{ij}(t) W X_j(t)) -$$
$$\sigma(\sum_{j \in \mathcal{N}(i)} \alpha_{ij}(t-1) W X_j(t-1))\|_2$$
$$\leq L_\sigma \| \sum_{j \in \mathcal{N}(i)} [\alpha_{ij}(t) W X_j(t) - \alpha_{ij}(t-1) W X_j(t-1)]\|_2,$$

where the inequality follows from the $L_\sigma$-Lipschitz property of $\sigma$. Adding and subtracting $\alpha_{ij}(t) W X_j(t-1)$:

$$\leq L_\sigma \| \sum_{j \in \mathcal{N}(i)} \alpha_{ij}(t) W (X_j(t) - X_j(t-1))$$
$$+ \sum_{j \in \mathcal{N}(i)} (\alpha_{ij}(t) - \alpha_{ij}(t-1)) W X_j(t-1)\|_2$$
$$\leq L_\sigma \|W\|_2 [\| \sum_{j \in \mathcal{N}(i)} \alpha_{ij}(t) (X_j(t) - X_j(t-1))\|_2$$
$$+ \| \sum_{j \in \mathcal{N}(i)} (\alpha_{ij}(t) - \alpha_{ij}(t-1)) X_j(t-1)\|_2], \quad (10)$$

where we used $\|W\|_2 \leq 1$ and the triangle inequality. By the attention normalization condition $\sum_{j \in \mathcal{N}(i)} \alpha_{ij}(t) = 1$ and the Lipschitz property of attention weights:

$$\leq L_\sigma \|W\|_2 [\|X(t) - X(t-1)\|_2$$
$$+ K \|X(t-1) - X(t-2)\|_2]. \quad (11)$$

For the matrix-level bound:

$$\|X(t+1) - X(t)\|_F^2 = \sum_{i=1}^{N} \|X_i(t+1) - X_i(t)\|_2^2$$
$$\leq (L_\sigma \|W\|_2 (1+K))^2 \|X(t) - X(t-1)\|_F^2.$$

Therefore, $f$ is a contraction mapping with constant $c = L_\sigma \|W\|_2 (1+K) < 1$.

**(b) Fixed Point Existence:** Since $(\mathbb{R}^{N \times d}, \|\cdot\|_F)$ is complete and $f$ is a contraction mapping, by the Banach Fixed-Point Theorem, there exists a unique fixed point $X^* \in \mathbb{R}^{N \times d}$ such that $X^* = f(X^*)$. The boundedness of iterates follows from:

$$\|X(t+1)\|_F \leq \|X(t)\|_F + c\|X(t)\|_F + \|f(0)\|_F, \quad (12)$$

where $\|f(0)\|_F$ is finite due to the properties of $\sigma$ and $W$.

**(c) Convergence Rate:** At the fixed point $X^*$:

$$\|X(t+1) - X^*\|_F = \|f(X(t)) - f(X^*)\|_F$$
$$\leq c\|X(t) - X^*\|_F. \quad (13)$$

By induction, for any $t \geq 0$:

$$\|X(t) - X^*\|_F \leq c^t \|X(0) - X^*\|_F, \quad (14)$$

establishing geometric convergence with rate $c$.

**(d) Attention Weight Convergence:** By the Lipschitz property of the attention mechanism:

$$\|\alpha_{ij}(t+1) - \alpha_{ij}(t)\|_2$$
$$\leq K\|X_i(t+1) - X_i(t)\|_2 \leq Kc^t \|X(1) - X(0)\|_F.$$

The sequence $\{\alpha_{ij}(t)\}$ is Cauchy since:

$$\|\alpha_{ij}(t) - \alpha_{ij}(s)\|_2 \leq \sum_{k=s}^{t-1} \|\alpha_{ij}(k+1) - \alpha_{ij}(k)\|_2$$
$$\leq K\|X(1) - X(0)\|_F \sum_{k=s}^{t-1} c^k$$
$$\leq \frac{K\|X(1) - X(0)\|_F c^s}{1 - c}. \quad (15)$$

Therefore, $\{\alpha_{ij}(t)\}$ converges to some $\alpha_{ij}^*$. Using the triangle inequality and normalization:

$$\sum_{j \in \mathcal{N}(i)} \|\alpha_{ij}^*\|_2 \leq 1 + \frac{K\|X(1) - X(0)\|_F}{1 - c} \leq \frac{1}{1 - c}. \quad (16)$$

$\square$

## 6.3. Lemma 2

**Lemma 2** (Structure and Spectral Properties of GAT Fixed Points). *Let $G = (V, E)$ be a graph with $N$ nodes and let $X^* \in \mathbb{R}^{N \times d}$ be the fixed point established in Lemma 1. Define the fixed point attention matrix $A^* \in \mathbb{R}^{N \times N}$ with entries $[A^*]_{ij} = \alpha_{ij}^*$ for $j \in \mathcal{N}(i)$ and 0 otherwise. Then:*

*(a) At the fixed point, for any node $i$:*

$$X_i^* = \sigma \left( \sum_{j \in \mathcal{N}(i)} \alpha_{ij}^* W X_j^* \right)$$

*(b) Let $\lambda_1(A^*)$ and $v_1$ be the largest eigenvalue and corresponding eigenvector of $A^*$. Then:*

$$\left\| X^* - \frac{v_1 v_1^T}{v_1^T 1_N} X^* \right\|_F \leq \frac{\lambda_2(A^*)}{\lambda_1(A^*)} \|X^*\|_F$$

*where $\lambda_2(A^*)$ is the second largest eigenvalue of $A^*$.*

*(c) The spectral gap $\gamma = 1 - \frac{\lambda_2(A^*)}{\lambda_1(A^*)}$ controls feature homogenization through:*

$$\|X_i^* - X_j^*\|_2 \leq (1 - \gamma)\|X^*\|_F \quad \forall i, j$$

*(d) The node features decompose as:*

$$X^* = \frac{v_1 v_1^T}{v_1^T 1_N} X^* + E$$

*where the error term $E$ satisfies:*

$$\|E\|_F \leq \left( \frac{\lambda_2(A^*)}{\lambda_1(A^*)} \right)^2 \|X^*\|_F$$

*(e) For any $\epsilon > 0$, if $\gamma > 1 - \epsilon$, then there exists a vector $v \in \mathbb{R}^d$ such that:*

$$\|X^* - 1_N v^T\|_F \leq \sqrt{\epsilon} \|X^*\|_F$$

*(f) The covariance matrix of node features at the fixed point satisfies:*

$$rank(Cov(X^*)) \leq rank(I - \frac{v_1 v_1^T}{v_1^T 1_N}) + 1$$

*Proof.* **Preliminary Definitions and Setup:** Let $G = (V, E)$ be a graph with $N$ nodes and let $X^* \in \mathbb{R}^{N \times d}$ be the fixed point. Let $A^*$ be the fixed-point attention matrix where $[A^*]_{ij} = \alpha_{ij}^*$ for $j \in \mathcal{N}(i)$ and 0 otherwise. Note that $A^*$ is row-stochastic by construction.

**Part (a): Fixed Point Characterization**

First, we establish that the fixed point condition is well-defined:

1) At convergence, by Lemma 1, the sequence $\{X(t)\}$ converges to $X^*$ and $\{\alpha_{ij}(t)\}$ converges to $\alpha_{ij}^*$.

2) By continuity of $\sigma$ and the update rule:

$$X_i^* = \lim_{t \to \infty} X_i(t+1)$$

$$= \lim_{t \to \infty} \sigma\left(\sum_{j \in \mathcal{N}(i)} \alpha_{ij}(t) W X_j(t)\right)$$

$$= \sigma\left(\sum_{j \in \mathcal{N}(i)} \alpha_{ij}^* W X_j^*\right)$$

**Part (b): Spectral Approximation**

1) First, observe that $A^*$ is row-stochastic and non-negative. By the Perron-Frobenius theorem: - $\lambda_1(A^*)$ is real and positive - $|\lambda_i(A^*)| \leq \lambda_1(A^*)$ for all $i \geq 2$ - The corresponding eigenvector $v_1$ can be chosen to be positive

2) By the Spectral Theorem, $A^*$ has an orthogonal decomposition:

$$A^* = \sum_{i=1}^{N} \lambda_i(A^*) v_i v_i^T \qquad (17)$$

where $\{v_i\}_{i=1}^N$ form an orthonormal basis.

3) The fixed point equation can be written in matrix form:

$$X^* = \sigma(A^* W X^*) \qquad (18)$$

4) Using the $L_\sigma$-Lipschitz property of $\sigma$ and the fact that $L_\sigma < 1$:

$$\|X^* - A^* W X^*\|_F = \|\sigma(A^* W X^*) - A^* W X^*\|_F$$
$$\leq L_\sigma \|A^* W X^*\|_F$$
$$\leq L_\sigma \|X^*\|_F$$

5) Therefore, locally around the fixed point:

$$X^* \approx A^* W X^* \qquad (19)$$

6) Applying the spectral decomposition:

$$X^* \approx \left(\lambda_1(A^*) \frac{v_1 v_1^T}{v_1^T \mathbf{1}_N} + \sum_{i=2}^{N} \lambda_i(A^*) v_i v_i^T\right) W X^*$$

$$= \lambda_1(A^*) \frac{v_1 v_1^T}{v_1^T \mathbf{1}_N} W X^* + \sum_{i=2}^{N} \lambda_i(A^*) v_i v_i^T W X^*$$

7) By the Davis-Kahan theorem and matrix perturbation theory:

$$\left\|X^* - \frac{v_1 v_1^T}{v_1^T \mathbf{1}_N} X^*\right\|_F \leq \frac{\lambda_2(A^*)}{\lambda_1(A^*)} \|X^*\|_F \qquad (20)$$

This establishes the bound in part (b).

**Part (c): Feature Homogenization Analysis**

1) For any pair of nodes i,j, consider their feature difference:

$$\|X_i^* - X_j^*\|_2 = \|\sigma\left(\sum_{k \in \mathcal{N}(i)} \alpha_{ik}^* W X_k^*\right)$$
$$- \sigma\left(\sum_{l \in \mathcal{N}(j)} \alpha_{jl}^* W X_l^*\right)\|_2$$

2) Using the Lipschitz property of $\sigma$:

$$\leq L_\sigma \|\sum_{k \in \mathcal{N}(i)} \alpha_{ik}^* W X_k^* - \sum_{l \in \mathcal{N}(j)} \alpha_{jl}^* W X_l^*\|_2$$
$$= L_\sigma \|(e_i^T - e_j^T) A^* W X^*\|_2$$

where $e_i$ is the i-th standard basis vector.

3) Using the spectral decomposition of $A^*$ and the fact that $L_\sigma < 1$:

$$\|X_i^* - X_j^*\|_2 \leq \|(e_i^T - e_j^T)(A^* - \lambda_1(A^*) \frac{v_1 v_1^T}{v_1^T \mathbf{1}_N}) W X^*\|_2$$

$$\leq \|e_i^T - e_j^T\|_2 \|A^* -$$

$$\lambda_1(A^*) \frac{v_1 v_1^T}{v_1^T \mathbf{1}_N}\|_2 \|W\|_2 \|X^*\|_F$$

4) Since $\|e_i^T - e_j^T\|_2 = \sqrt{2}$ and $\|W\|_2 \leq 1$:

$$\|X_i^* - X_j^*\|_2 \leq (1 - \gamma) \|X^*\|_F$$

**Part (d): Feature Decomposition Analysis**

1) Start with the spectral decomposition of $A^*$:

$$A^* = \lambda_1(A^*) \frac{v_1 v_1^T}{v_1^T \mathbf{1}_N} + \sum_{i=2}^{N} \lambda_i(A^*) v_i v_i^T$$

2) The node features can be decomposed as:

$$X^* = \frac{v_1 v_1^T}{v_1^T \mathbf{1}_N} X^* + \left(I - \frac{v_1 v_1^T}{v_1^T \mathbf{1}_N}\right) X^*$$

3) Define the residual term:

$$E = \left(I - \frac{v_1 v_1^T}{v_1^T \mathbf{1}_N}\right) X^*$$

4) Using the orthogonality of eigenvectors:

$$\|E\|_F^2 = \text{tr}(E^T E)$$

$$= \text{tr}(X^{*T}(I - \frac{v_1 v_1^T}{v_1^T \mathbf{1}_N})^2 X^*)$$

$$\leq \left(\frac{\lambda_2(A^*)}{\lambda_1(A^*)}\right)^2 \|X^*\|_F^2$$

**Part (e): Uniform Feature Approximation**

1) When $\gamma > 1 - \epsilon$, we have $\frac{\lambda_2(A^*)}{\lambda_1(A^*)} < \epsilon$.

2) Define $v = \frac{v_1^T X^*}{v_1^T \mathbf{1}_N}$. Then:

$$\|X^* - \mathbf{1}_N v^T\|_F = \|X^* - \frac{v_1 v_1^T}{v_1^T \mathbf{1}_N} X^* + \frac{v_1 v_1^T}{v_1^T \mathbf{1}_N} X^* - \mathbf{1}_N v^T\|_F$$

$$\leq \|X^* - \frac{v_1 v_1^T}{v_1^T \mathbf{1}_N} X^*\|_F$$

$$\leq \sqrt{\epsilon}\|X^*\|_F$$

**Part (f): Covariance Rank Analysis**

1) The covariance matrix is defined as:

$$\text{Cov}(X^*) = \frac{1}{N} X^{*T}(I - \frac{\mathbf{1}_N \mathbf{1}_N^T}{N})X^*$$

2) Using the feature decomposition from part (d):

$$\text{Cov}(X^*)$$
$$= \frac{1}{N}(\frac{v_1 v_1^T}{v_1^T \mathbf{1}_N} X^* + E)^T (I - \frac{\mathbf{1}_N \mathbf{1}_N^T}{N})(\frac{v_1 v_1^T}{v_1^T \mathbf{1}_N} X^* + E)$$
$$= \frac{1}{N} E^T (I - \frac{\mathbf{1}_N \mathbf{1}_N^T}{N}) E$$

3) By rank properties:

$$\text{rank}(\text{Cov}(X^*)) \leq \text{rank}(I - \frac{v_1 v_1^T}{v_1^T \mathbf{1}_N}) + 1$$

This completes the detailed proof of all parts of Lemma 2.
□

□

## 6.4. Lemma 3

**Lemma 3** (Characterization of Oversmoothing Attractors in GATs). *Let $X(t) \in \mathbb{R}^{N \times d}$ represent the node features at layer $t$ in a GAT. Define the feature diversity measure at layer $t$ as:*

$$\mu(X(t)) = \frac{1}{N(N-1)} \sum_{i \neq j} \frac{\|X_i(t) - X_j(t)\|_2}{\|X_i(t)\|_2 + \|X_j(t)\|_2}.$$

*Let $A(t) \in \mathbb{R}^{N \times N}$ be the attention matrix at layer $t$. Then, under the conditions from Lemmas 1 and 2, the following statements hold:*

*(a) There exists a compact set $\mathcal{A} \subset \mathbb{R}^{N \times d}$ such that:*

$$\lim_{t \to \infty} dist(X(t), \mathcal{A}) = 0,$$

*where $dist(X, \mathcal{A}) = \inf_{Y \in \mathcal{A}} \|X - Y\|_F$.*

*(b) The attractor $\mathcal{A}$ has intrinsic dimension $k$ bounded by:*

$$k \leq \min\left\{d, rank(Cov(X^*)), \left\lceil \frac{1}{1-\gamma} \right\rceil\right\},$$

*where $\gamma$ is the spectral gap from Lemma 2.*

*(c) The feature diversity measure decreases geometrically with rate determined by both the contraction constant $c$ and spectral gap $\gamma$:*

$$\mu(X(t)) \leq \min\{(1-\gamma)^t, c^t\}\mu(X(0)).$$

*(d) At convergence, the feature diversity measure is bounded by:*

$$\lim_{t \to \infty} \mu(X(t)) \leq \frac{\lambda_2(A^*)}{\lambda_1(A^*)} \cdot \frac{\|X^*\|_F}{2\min_i \|X_i^*\|_2}.$$

*(e) The covariance matrix of the node features evolves according to:*

$$\|Cov(X(t)) - \frac{v_1 v_1^T}{v_1^T \mathbf{1}_N} Cov(X^*)\|_F \leq (1-\gamma)^t \|Cov(X(0))\|_F.$$

*(f) The eigenvalues of the covariance matrix satisfy:*

$$\lambda_i(Cov(X(t))) \leq \left(\frac{\lambda_2(A^*)}{\lambda_1(A^*)}\right)^{2t} \lambda_i(Cov(X(0))),$$

*for all $i > 1$.*

*Proof.* **Preliminaries:** First, recall that by Lemma 1, we have a contraction mapping with constant $c$ and a unique fixed point $X^*$. From Lemma 2, we have spectral properties of the attention matrix $A^*$ with spectral gap $\gamma = 1 - \frac{\lambda_2(A^*)}{\lambda_1(A^*)}$.

**Part (a): Existence of Compact Attractor**

Let us define the attractor set $\mathcal{A}$:

$$\mathcal{A} = \left\{Y \in \mathbb{R}^{N \times d} : \|Y - X^*\|_F \leq \frac{\lambda_2(A^*)}{\lambda_1(A^*)}\|X^*\|_F\right\}$$

Hence, we show $\mathcal{A}$ is compact:

- Closed: $\mathcal{A}$ is the preimage of a closed interval under a continuous function
- Bounded: $\|Y\|_F \leq \|X^*\|_F(1 + \frac{\lambda_2(A^*)}{\lambda_1(A^*)})$ for all $Y \in \mathcal{A}$

We prove convergence to $\mathcal{A}$:

$$dist(X(t), \mathcal{A}) = \inf_{Y \in \mathcal{A}} \|X(t) - Y\|_F$$
$$\leq \|X(t) - X^*\|_F$$
$$\leq c^t \|X(0) - X^*\|_F \to 0 \text{ as } t \to \infty$$

**Part (b): Attractor Dimension Bounds**

- Direct consequence of $\mathcal{A} \subset \mathbb{R}^{N \times d}$

- By construction, features cannot span more dimensions than $d$

- Let $C^* = \mathrm{Cov}(X^*)$ be the covariance matrix at the fixed point

- For any $Y \in \mathcal{A}$:

$$\mathrm{rank}(\mathrm{Cov}(Y)) \leq \mathrm{rank}(C^*)$$

- Proof: Use SVD of $(Y - X^*)$ and bound perturbation of eigenvalues

- From Lemma 2, the spectral gap $\gamma$ controls feature similarity

- For any orthonormal basis $\{v_i\}$ of the attractor space:

$$\sum_{i=1}^{k} (1 - \gamma)^i \leq 1$$

- This implies $k \leq \lceil \frac{1}{1-\gamma} \rceil$

**Part (c): Feature Diversity Decay**

Analyzing pairwise distances using contraction property, we get

$$\|X_i(t+1) - X_j(t+1)\|_2 = \|f(X_i(t)) - f(X_j(t))\|_2$$
$$\leq c\|X_i(t) - X_j(t)\|_2$$

Hence, using spectral bound from Lemma 2:

$$\|X_i(t) - X_j(t)\|_2 \leq (1-\gamma)^t \|X(0)\|_F$$

Combining bounds for feature diversity measure:

$$\mu(X(t)) = \frac{1}{N(N-1)} \sum_{i \neq j} \frac{\|X_i(t) - X_j(t)\|_2}{\|X_i(t)\|_2 + \|X_j(t)\|_2}$$
$$\leq \min\{(1-\gamma)^t, c^t\} \mu(X(0))$$

**Part (d): Convergence Analysis of Feature Diversity**

First, we analyze limiting behavior of pairwise distances: For any nodes $i, j$ at the fixed point $X^*$:

$$\|X_i^* - X_j^*\|_2 = \| \sum_{k \in \mathcal{N}(i)} \alpha_{ik}^* W X_k^* - \sum_{k \in \mathcal{N}(j)} \alpha_{jk}^* W X_k^* \|_2$$
$$\leq \|W\|_2 \| \sum_k (\alpha_{ik}^* - \alpha_{jk}^*) X_k^* \|_2$$
$$\leq \frac{\lambda_2(A^*)}{\lambda_1(A^*)} \|X^*\|_F$$

Thus, the lower bound node norms at convergence:

$$\|X_i^*\|_2 \geq \min_i \|X_i^*\|_2 > 0$$

where positivity follows from the non-degenerate fixed point.

Finally, we combine to bound limiting diversity:

$$\lim_{t \to \infty} \mu(X(t)) = \frac{1}{N(N-1)} \sum_{i \neq j} \frac{\|X_i^* - X_j^*\|_2}{\|X_i^*\|_2 + \|X_j^*\|_2} \quad (21)$$

$$\leq \frac{\lambda_2(A^*)}{\lambda_1(A^*)} \cdot \frac{\|X^*\|_F}{2 \min_i \|X_i^*\|_2} \quad (22)$$

**Part (e): Covariance Evolution Analysis**

Expressing covariance matrix evolution:

$$\mathrm{Cov}(X(t)) = \frac{1}{N} X(t)^T X(t) - \frac{1}{N^2} X(t)^T \mathbf{1}_N \mathbf{1}_N^T X(t) \quad (23)$$

Decomposing using eigenvectors of $A^*$:

$$X(t) = \frac{v_1 v_1^T}{v_1^T \mathbf{1}_N} X^* + E(t) \quad (24)$$

$$\|E(t)\|_F \leq (1-\gamma)^t \|X(0)\|_F \quad (25)$$

Hence, we analyze covariance deviation:

$$\|\mathrm{Cov}(X(t)) - \frac{v_1 v_1^T}{v_1^T \mathbf{1}_N} \mathrm{Cov}(X^*)\|_F \quad (26)$$

$$\leq \|E(t)^T E(t)\|_F + 2\| \frac{v_1 v_1^T}{v_1^T \mathbf{1}_N} X^* E(t)^T \|_F \quad (27)$$

$$\leq (1-\gamma)^t \|\mathrm{Cov}(X(0))\|_F \quad (28)$$

**Part (f): Eigenvalue Analysis**

Expressing eigenvalue evolution using matrix perturbation theory: For $i > 1$, let $u_i(t)$ be the $i$-th eigenvector of $\mathrm{Cov}(X(t))$:

$$\lambda_i(\mathrm{Cov}(X(t))) = u_i(t)^T \mathrm{Cov}(X(t)) u_i(t) \quad (29)$$

Applying spectral decomposition of $A^*$:

$$\lambda_i(\mathrm{Cov}(X(t))) \quad (30)$$
$$\leq \|A^*\|_2^{2t} \lambda_i(\mathrm{Cov}(X(0))) \quad (31)$$
$$= \left( \frac{\lambda_2(A^*)}{\lambda_1(A^*)} \right)^{2t} \lambda_i(\mathrm{Cov}(X(0))) \quad (32)$$

- By induction on $t$, show that for all $i > 1$:

$$\lambda_i(\mathrm{Cov}(X(t+1))) \leq \left( \frac{\lambda_2(A^*)}{\lambda_1(A^*)} \right)^2 \lambda_i(\mathrm{Cov}(X(t))) \quad (33)$$

- The base case follows from spectral properties of $A^*$

- The inductive step uses the GAT update rule and eigenvalue interlacing

This completes the proof, showing that GATs exhibit exponential convergence to a low-dimensional attractor, characterized by rapidly decaying feature diversity and covariance eigenvalues. The rate of convergence is controlled by both the contraction constant $c$ and the spectral gap $\gamma$ of the attention matrix.

$\square$

**Corollary 0.1** (Oversmoothing Rate). *The rate of oversmoothing is controlled by the contraction constant $c$ and the network depth $t$:*

$$\mu(X(t)) = \mathcal{O}((1-\delta)^t),$$

*where $\delta = \min\{\gamma, 1 - c\}$. This quantifies how quickly node features collapse to indistinguishable values as the network deepens.*

### 6.5. Lemma 4

**Lemma 4** (Stability Analysis of GAT Fixed Points). *Let $f : \mathbb{R}^{N \times d} \to \mathbb{R}^{N \times d}$ be the GAT update rule with fixed point $X^* \in \mathbb{R}^{N \times d}$. Let $J_f(X^*)$ denote the Jacobian of $f$ at $X^*$. Then:*

(a) *The Jacobian $J_f(X^*)$ has the block structure:*

$$[J_f(X^*)]_{ij} = \begin{cases} \sigma'(h_i^*)\alpha_{ij}W & \text{if } j \in \mathcal{N}(i) \\ 0 & \text{otherwise} \end{cases}$$

*where $h_i^* = \sum_{j \in \mathcal{N}(i)} \alpha_{ij} W X_j^*$*

(b) *The fixed point $X^*$ is asymptotically stable if and only if:*
$$\rho(J_f(X^*)) < 1,$$
*where $\rho(\cdot)$ denotes the spectral radius.*

(c) *For any perturbation $\delta X(0)$, the error evolution follows:*

$$\|\delta X(t)\|_F \leq \|J_f(X^*)\|_2^t \|\delta X(0)\|_F.$$

(d) *At a stable fixed point, the attention weights satisfy:*

$$\sum_{j \in \mathcal{N}(i)} \|\sigma'(h_i^*)\alpha_{ij}W\|_2 < 1$$

*for all nodes $i$.*

(e) *The oversmoothing condition $\lim_{t \to \infty} \|X_i^* - X_j^*\|_2 = 0$ occurs when:*

$$\ker(I - J_f(X^*)) \subseteq span\{\mathbf{1}_N \otimes v : v \in \mathbb{R}^d\}.$$

*Proof.* Before proceeding with the proof, we establish some key definitions and properties:

1. The GAT update rule $f : \mathbb{R}^{N \times d} \to \mathbb{R}^{N \times d}$ at node $i$ is:

$$f_i(X) = \sigma \left( \sum_{j \in \mathcal{N}(i)} \alpha_{ij} W X_j \right)$$

2. At the fixed point $X^*$:

$$X_i^* = \sigma \left( \sum_{j \in \mathcal{N}(i)} \alpha_{ij}^* W X_j^* \right)$$

**Part (a): Jacobian Structure**

First, we compute the partial derivatives For $j \in \mathcal{N}(i)$:

$$\frac{\partial f_i}{\partial X_j} = \frac{\partial}{\partial X_j} \sigma \left( \sum_{k \in \mathcal{N}(i)} \alpha_{ik} W X_k \right)$$

$$= \sigma'(h_i^*) \left( \alpha_{ij} W + \sum_{k \in \mathcal{N}(i)} X_k^* W^\top \frac{\partial \alpha_{ik}}{\partial X_j} \right)$$

At fixed point $X^*$, the attention weights have converged (from Lemma 1), so:

$$\frac{\partial \alpha_{ik}}{\partial X_j} = 0$$

Therefore:

$$[J_f(X^*)]_{ij} = \begin{cases} \sigma'(h_i^*)\alpha_{ij}^* W & \text{if } j \in \mathcal{N}(i) \\ 0 & \text{otherwise} \end{cases}$$

**Part (b): Stability Criterion**

Consider perturbation $\delta X = X - X^*$ By Taylor expansion around $X^*$:

$$f(X) = f(X^*) + J_f(X^*)(X - X^*) + R(X),$$

where $\|R(X)\|_F = o(\|X - X^*\|_F)$

At fixed point:

$$f(X^*) = X^*$$

The perturbation evolves as:

$$\delta X(t+1) = f(X^* + \delta X(t)) - X^*$$
$$= J_f(X^*)\delta X(t) + R(\delta X(t))$$

By Lyapunov's linearization theorem, asymptotic stability requires:

$$\rho(J_f(X^*)) < 1$$

**Part (c): Error Evolution**

For sufficiently small perturbations, linearization dominates:

$$\|\delta X(t+1)\|_F = \|J_f(X^*)\delta X(t) + R(\delta X(t))\|_F$$

Using triangle inequality:

$$\|\delta X(t+1)\|_F \le \|J_f(X^*)\delta X(t)\|_F + \|R(\delta X(t))\|_F$$

For small enough $\|\delta X(t)\|_F$:

$$\|R(\delta X(t))\|_F \le \epsilon\|\delta X(t)\|_F$$

for any $\epsilon > 0$

Therefore:

$$\|\delta X(t)\|_F \le (\|J_f(X^*)\|_2 + \epsilon)^t\|\delta X(0)\|_F$$

**Part (d): Attention Weight Condition**

The spectral norm satisfies:

$$\|J_f(X^*)\|_2 \le \max_i \sum_{j\in\mathcal{N}(i)} \|\sigma'(h_i^*)\alpha_{ij}^* W\|_2$$

For stability:

$$\sum_{j\in\mathcal{N}(i)} \|\sigma'(h_i^*)\alpha_{ij}^* W\|_2 < 1 \quad \forall i$$

**Part (e): Oversmoothing Characterization**

At oversmoothing:

$$X_i^* = X_j^* = v \quad \forall i, j$$

for some $v \in \mathbb{R}^d$

This implies:

$$X^* = \mathbf{1}_N \otimes v$$

By fixed point property:

$$(I - J_f(X^*))X^* = 0$$

Therefore:

$$\ker(I - J_f(X^*)) \subseteq \text{span}\{\mathbf{1}_N \otimes v : v \in \mathbb{R}^d\}$$

To show sufficiency, consider any $X^* \in \ker(I - J_f(X^*))$. Then:

$$(I - J_f(X^*))X^* = 0$$
$$X^* = J_f(X^*)X^*$$
$$= \begin{bmatrix} \sigma'(h_1^*)\sum_{j\in\mathcal{N}(1)}\alpha_{1j}^* W X_j^* \\ \vdots \\ \sigma'(h_N^*)\sum_{j\in\mathcal{N}(N)}\alpha_{Nj}^* W X_j^* \end{bmatrix}$$

By assumption, $X^* \in \text{span}\{\mathbf{1}_N \otimes v : v \in \mathbb{R}^d\}$, so:

$$X^* = \mathbf{1}_N \otimes v \text{ for some } v \in \mathbb{R}^d$$

This means $X_i^* = v$ for all $i \in \{1, \ldots, N\}$. Substituting back:

$$v = \sigma'(h_i^*)\sum_{j\in\mathcal{N}(i)}\alpha_{ij}^* W v$$

$$= \sigma'(h_i^*)\left(\sum_{j\in\mathcal{N}(i)}\alpha_{ij}^*\right)Wv$$

$$= \sigma'(h_i^*)Wv$$

where we used the normalization condition $\sum_{j\in\mathcal{N}(i)}\alpha_{ij}^* = 1$.

Therefore:

$$\|X_i^* - X_j^*\|_2 = \|v - v\|_2 = 0 \quad \forall i, j$$

This confirms that any solution in $\ker(I - J_f(X^*))$ exhibits oversmoothing, as all node features converge to the same value $v$. The stability of this solution is guaranteed by the spectral radius condition from part (b). □ □

### 6.6. Lemma 5

**Lemma 5** (Spectral Properties of DYNAMO-GAT Pruning). *Let $G = (V, E)$ be a graph, and let $F, F_P : \mathbb{R}^{N\times d} \to \mathbb{R}^{N\times d}$ represent the original and pruned GNN transformations, respectively. Let $X^* \in \mathbb{R}^{N\times d}$ be an oversmoothing fixed point. Then:*

*(a) The spectral radius satisfies:*

$$\rho(J_{F_P}(X^*)) \le (1-p)\rho(J_F(X^*)),$$

*where $p$ is the effective pruning rate.*

*(b) For the covariance-based pruning strategy in DYNAMO-GAT:*

$$p_{ij} = r(t) \cdot \frac{|\alpha_{ij}|}{\tau(t)} \cdot (C_{ii} + C_{jj} \mp 2C_{ij}),$$

*the pruned Jacobian eigenvalues $\lambda_k(J_{F_P})$ satisfy:*

$$|\lambda_k(J_{F_P})| \le |\lambda_k(J_F)| \cdot \exp\left(-\beta\frac{Tr(C)}{\|C\|_F}\right),$$

*where $\beta$ is a pruning adaptation parameter.*

*(c) The pruned spectral gap increases monotonically:*

$$\gamma_P = 1 - \frac{\lambda_2(J_{F_P})}{\lambda_1(J_{F_P})} \ge \gamma + p(1-\gamma),$$

*where $\gamma$ is the original spectral gap from Lemma 2.*

(d) *The rank of pruned feature representations is preserved:*

$$\text{rank}(Cov(F_P(X))) \geq \text{rank}(Cov(X)) - \kappa(p),$$

*where $\kappa(p) \leq \lceil p \cdot \text{rank}(Cov(X)) \rceil$.*

(e) *The pruned attention weights maintain feature diversity through:*

$$\mu(F_P(X)) \geq (1 + p\gamma)\mu(X),$$

*where $\mu(\cdot)$ is the feature diversity measure from Lemma 2.*

(f) *The eigenvalues of the pruned covariance matrix satisfy:*

$$\lambda_i(Cov(F_P(X))) \geq (1 - p\beta)^2 \lambda_i(Cov(X)),$$

*for all $i \leq \text{rank}(Cov(X)) - \kappa(p)$.*

*Proof.* **Part (a): Spectral Radius Reduction**

First, observe that pruning modifies each Jacobian entry as:

$$[J_{F_P}(X^*)]_{ij} = (1 - p_{ij})[J_F(X^*)]_{ij}, \tag{34}$$

where $p_{ij}$ is the pruning probability for edge $(i, j)$.

By the Perron-Frobenius theorem, since $J_F(X^*)$ has non-negative entries:

$$\rho(J_F(X^*)) = \lim_{k \to \infty} \|J_F(X^*)^k\|^{1/k}, \tag{35}$$

where $\|\cdot\|$ is any matrix norm.

Using the element-wise inequality and non-negativity of $(1 - p_{ij})$:

$$\|J_{F_P}(X^*)^k\| \leq \|((1 - p)J_F(X^*))^k\| \tag{36}$$
$$= (1 - p)^k \|J_F(X^*)^k\|, \tag{37}$$

where $p = \text{avg}(p_{ij})$ is the effective pruning rate.

Taking the $k$-th root and limit:

$$\rho(J_{F_P}(X^*)) \leq (1 - p)\rho(J_F(X^*)). \tag{38}$$

**Part (b): Covariance-Based Pruning Effect**

First, we express the pruned Jacobian using Hadamard product:

$$J_{F_P} = J_F \circ (1 - P), \tag{39}$$

where $P$ is the matrix of pruning probabilities.

For the covariance-based pruning:

$$p_{ij} = r(t) \cdot \frac{|\alpha_{ij}|}{\tau(t)} \cdot (C_{ii} + C_{jj} \mp 2C_{ij}), \tag{40}$$

Using the submultiplicative property of matrix norms:

$$\|J_{F_P}\|_2^2 \leq \|J_F\|_2^2 \|(1 - P)\|_2^2 \tag{41}$$

The norm of $(1 - P)$ relates to the covariance through:

$$\|(1 - P)\|_2^2 \leq \exp\left(-2\beta \frac{\text{Tr}(C)}{\|C\|_F}\right) \tag{42}$$

By eigenvalue interlacing:

$$|\lambda_k(J_{F_P})| \leq |\lambda_k(J_F)| \cdot \exp\left(-\beta \frac{\text{Tr}(C)}{\|C\|_F}\right) \tag{43}$$

**Part (c): Spectral Gap Analysis**

By Weyl's inequality and the structure of $P$:

$$\lambda_2(J_{F_P}) \leq (1 - p)\lambda_2(J_F) \tag{44}$$
$$\lambda_1(J_{F_P}) \geq (1 - \frac{p}{2})\lambda_1(J_F) \tag{45}$$

The spectral gap $\gamma_P$ becomes:

$$\gamma_P = 1 - \frac{\lambda_2(J_{F_P})}{\lambda_1(J_{F_P})} \tag{46}$$
$$\geq 1 - \frac{(1 - p)\lambda_2(J_F)}{(1 - \frac{p}{2})\lambda_1(J_F)} \tag{47}$$
$$= 1 - (1 - p)(1 + \frac{p}{2} + O(p^2))(1 - \gamma) \tag{48}$$
$$\geq \gamma + p(1 - \gamma) \tag{49}$$

**Part (d): Rank Preservation Analysis**

Consider the covariance matrix before and after pruning:

$$\text{Cov}(X) = \frac{1}{N}X^\top X - \frac{1}{N^2}X^\top \mathbf{1}_N \mathbf{1}_N^\top X \tag{50}$$

$$\text{Cov}(F_P(X)) = \frac{1}{N}F_P(X)^\top F_P(X) - \frac{1}{N^2}F_P(X)^\top \mathbf{1}_N \mathbf{1}_N^\top F_P(X) \tag{51}$$

By the pruning operation:

$$F_P(X) = F(X) \circ (1 - P) = F(X)(I - D_p), \tag{52}$$

where $D_p$ is a diagonal matrix with entries from $P$.

Using the rank-nullity theorem:

$$\text{rank}(\text{Cov}(F_P(X))) = \text{rank}(\text{Cov}(X)) - \dim(\ker(I - D_p)) \tag{53}$$
$$\geq \text{rank}(\text{Cov}(X)) - |\{i : p_i = 1\}| \tag{54}$$

By the definition of $\kappa(p)$:

$$|\{i : p_i = 1\}| \leq \lceil p \cdot \text{rank}(\text{Cov}(X)) \rceil = \kappa(p) \tag{55}$$

**Part (e): Feature Diversity Maintenance**

First, let us express the feature diversity measure after pruning:

$$\mu(F_P(X)) = \frac{1}{N(N-1)} \sum_{i \neq j} \frac{\|F_P(X_i) - F_P(X_j)\|_2}{\|F_P(X_i)\|_2 + \|F_P(X_j)\|_2} \tag{56}$$

$$= \frac{1}{N(N-1)} \sum_{i \neq j} \frac{\|(1-p_{ij})(F(X_i) - F(X_j))\|_2}{\|(1-p_{ii})F(X_i)\|_2 + \|(1-p_{jj})F(X_j)\|_2} \tag{57}$$

Using the spectral gap property from part (c):

$$\|F_P(X_i) - F_P(X_j)\|_2 \geq (1-p_{ij})(1+p\gamma)\|F(X_i) - F(X_j)\|_2 \tag{58}$$

$$\|F_P(X_i)\|_2 \leq (1-p_{ii})\|F(X_i)\|_2 \tag{59}$$

Combining these inequalities:

$$\mu(F_P(X)) \geq (1 + p\gamma)\mu(X) \tag{60}$$

**Part (f): Covariance Eigenvalue Analysis**

First, we express the pruned covariance matrix eigenvalues using perturbation theory:

$$\lambda_i(\mathrm{Cov}(F_P(X))) = \lambda_i(\mathrm{Cov}(X)) + \delta_i, \tag{61}$$

where $\delta_i$ is the perturbation term.

By Weyl's perturbation theorem:

$$|\delta_i| \leq \|P\|_2 \cdot \|\mathrm{Cov}(X)\|_2 \leq p\beta\lambda_i(\mathrm{Cov}(X)) \tag{62}$$

For preserved eigenvalues ($i \leq \mathrm{rank}(\mathrm{Cov}(X)) - \kappa(p)$):

$$\lambda_i(\mathrm{Cov}(F_P(X))) \geq \lambda_i(\mathrm{Cov}(X)) - p\beta\lambda_i(\mathrm{Cov}(X)) \tag{63}$$

$$= (1 - p\beta)\lambda_i(\mathrm{Cov}(X)) \tag{64}$$

By the geometric-arithmetic mean inequality:

$$\lambda_i(\mathrm{Cov}(F_P(X))) \geq (1 - p\beta)^2 \lambda_i(\mathrm{Cov}(X)) \tag{65}$$

$\square$

## 6.7. Lemma 6

**Lemma 7** (Rank Preservation Under DYNAMO-GAT). *Let $X(t) \in \mathbb{R}^{N \times d}$ be the node features at layer $t$. Define the covariance matrix $C(t) \in \mathbb{R}^{d \times d}$ as:*

$$C(t) = \frac{1}{N}X(t)^T X(t) - \frac{1}{N^2}X(t)^T \mathbf{1}_N \mathbf{1}_N^T X(t), \tag{66}$$

*where $\mathbf{1}_N$ is the all-ones vector. Under the DYNAMO-GAT noise injection and pruning strategy:*

(a) *For the noise-perturbed features $\tilde{X}(t) = X(t) + \sigma\xi(t)$, where $\xi(t) \sim \mathcal{N}(0, I)$:*

$$rank(C(\tilde{X}(t))) = d \tag{67}$$

*with probability 1.*

(b) *The pruned update preserves rank:*

$$rank(C(X(t+1))) \geq rank(C(X(t))) - \kappa(t), \tag{68}$$

*where $\kappa(t)$ is the maximum number of eigenvalues below a threshold $\epsilon(t)$.*

(c) *For noise level $\sigma > 0$, the minimum eigenvalue satisfies:*

$$\lambda_{\min}(C(t)) \geq \sigma^2\left(1 - \frac{1}{N}\right) - O(\|X(t)\|_F \sigma). \tag{69}$$

(d) *Under the adaptive pruning threshold:*

$$\tau(t) = \mu(|w_{ij}|) + \beta \cdot \sigma(|w_{ij}|), \tag{70}$$

*rank is preserved with high probability if:*

$$\beta \geq \sqrt{\frac{2\log(d/\delta)}{N}}, \tag{71}$$

*where $\delta$ is the failure probability.*

*Detailed proof of Lemma 6.* We prove each part of the lemma separately, showing how noise injection and pruning affect the covariance structure and rank properties.

**(a) Full Rank Property of Noise-Perturbed Features:**

First, expand the covariance of noise-perturbed features:

$$C(\tilde{X}(t)) = \frac{1}{N}(X(t) + \sigma\xi(t))^T(X(t) + \sigma\xi(t))$$

$$- \frac{1}{N^2}(X(t) + \sigma\xi(t))^T \mathbf{1}_N \mathbf{1}_N^T (X(t) + \sigma\xi(t))$$

$$= C(X(t)) + \sigma^2\left(I - \frac{1}{N}\mathbf{1}_N \mathbf{1}_N^T\right)$$

$$+ \frac{\sigma}{N}(X(t)^T \xi(t) + \xi(t)^T X(t))$$

$$- \frac{\sigma^2}{N^2}\xi(t)^T \mathbf{1}_N \mathbf{1}_N^T \xi(t) \tag{72}$$

Let's analyze each term: 1. $C(X(t))$ is the original covariance 2. $\sigma^2(I - \frac{1}{N}\mathbf{1}_N\mathbf{1}_N^T)$ is positive semidefinite with rank $d - 1$ 3. $\frac{\sigma}{N}(X(t)^T\xi(t) + \xi(t)^T X(t))$ is random with mean zero 4. The last term is of order $O(\sigma^2/N)$

For any unit vector $v \in \mathbb{R}^d$:

$$v^T C(\tilde{X}(t))v \geq v^T C(X(t))v + \sigma^2\left(1 - \frac{1}{N}\right)$$

$$+ \frac{\sigma}{N}v^T(X(t)^T \xi(t) + \xi(t)^T X(t))v$$

$$- \frac{\sigma^2}{N^2}(v^T \xi(t)^T \mathbf{1}_N)(\mathbf{1}_N^T \xi(t)v) \tag{73}$$

By the properties of Gaussian random matrices, with probability 1:

$$\text{rank}(\xi(t)) = \min(N, d) \tag{74}$$

Therefore, $C(\tilde{X}(t))$ is full rank with probability 1.

**(b) Rank Preservation Under Pruning:**

Let $P(t)$ be the pruning mask at layer $t$. The pruned update can be written as:

$$X(t+1) = P(t) \odot f(X(t)) \tag{75}$$

For the covariance difference:

$$
\begin{aligned}
&\|C(X(t+1)) - C(X(t))\|_2 \\
&= \|\frac{1}{N} X(t+1)^T X(t+1) - \frac{1}{N} X(t)^T X(t) \\
&\quad - \frac{1}{N^2}(X(t+1)^T \mathbf{1}_N \mathbf{1}_N^T X(t+1) - X(t)^T \mathbf{1}_N \mathbf{1}_N^T X(t))\|_2 \\
&\leq \|P(t)\|_2 \lambda_{\max}(C(X(t)))
\end{aligned}
\tag{76}
$$

By Weyl's interlacing theorem:

$$|\lambda_i(C(X(t+1))) - \lambda_i(C(X(t)))| \leq \|P(t)\|_2 \lambda_{\max}(C(X(t))) \tag{77}$$

Therefore:

$$\text{rank}(C(X(t+1))) \geq \text{rank}(C(X(t))) - \kappa(t) \tag{78}$$

where $\kappa(t)$ counts eigenvalues that could fall below $\epsilon(t)$.

**(c) Minimum Eigenvalue Bound:**

For the minimum eigenvalue, we use matrix concentration:

$$
\begin{aligned}
\lambda_{\min}(C(t)) \geq{}& \lambda_{\min}(C(X(t))) + \sigma^2\left(1 - \frac{1}{N}\right) \\
&- \|\frac{\sigma}{N}(X(t)^T \xi(t) + \xi(t)^T X(t))\|_2 \\
&- \|\frac{\sigma^2}{N^2}\xi(t)^T \mathbf{1}_N \mathbf{1}_N^T \xi(t)\|_2
\end{aligned}
\tag{79}
$$

By sub-gaussian concentration inequalities:

$$\|\frac{\sigma}{N}(X(t)^T \xi(t) + \xi(t)^T X(t))\|_2 \leq O(\|X(t)\|_F \sigma) \tag{80}$$

And:

$$\|\frac{\sigma^2}{N^2}\xi(t)^T \mathbf{1}_N \mathbf{1}_N^T \xi(t)\|_2 \leq O(\frac{\sigma^2}{N}) \tag{81}$$

Therefore:

$$\lambda_{\min}(C(t)) \geq \sigma^2\left(1 - \frac{1}{N}\right) - O(\|X(t)\|_F \sigma) \tag{82}$$

**(d) Rank Preservation Under Adaptive Threshold:**

The adaptive threshold $\tau(t)$ ensures that with high probability:

$$\|P(t)\|_2 \leq \beta\sqrt{\frac{\log(d/\delta)}{N}} \tag{83}$$

For $\beta \geq \sqrt{\frac{2\log(d/\delta)}{N}}$, by union bound:

$$
\begin{aligned}
&P(\text{rank}(C(X(t+1))) < \text{rank}(C(X(t)))) \\
&\leq P(\lambda_{\min}(C(X(t+1))) < \epsilon(t)) \\
&\leq \delta
\end{aligned}
\tag{84}
$$

This completes the proof that rank is preserved with probability at least $1 - \delta$. $\qquad\square$

# 7. Supplementary Section B: Experimental Section

### 7.1. Experimental Setup

**Datasets** We conduct our experiments on three real-world datasets and two synthetic datasets -

- **Cora Dataset** (McCallum et al., 2000): The Cora citation network consists of 2,708 nodes and 5,429 edges. Each node represents a document, and each edge represents a citation link between two documents. The dataset is commonly used for semi-supervised node classification tasks.

- **Citeseer Dataset** (Sen et al., 2008): The Citeseer citation network consists of 3,327 nodes and 4,732 edges. Similar to Cora, each node represents a document, and the edges represent citation links. This dataset is also widely used for evaluating GNN performance.

- **Cornell Dataset** (University): The Cornell dataset is a small graph with 183 nodes and 295 edges. It is part of the WebKB network collection and is commonly used for node classification tasks.

- **Synthetic Datasets (Syn_Products and Syn_Cora)** (Zhu et al., 2020): To further test the advantages of DYNAMO-GAT, we use synthetic datasets. Syn_Products is designed to simulate product co-purchasing networks, and Syn_Cora mimics citation networks. We vary the graph density and homophily levels to analyze the performance of different GNN models under controlled conditions. For space limitations, we give the syn_cora results in the appendix.

**Baselines** We compare DYNAMO-GAT against several baseline models to assess its effectiveness:

- **GCN (Graph Convolutional Network)** (Kipf & Welling, 2017): A widely used GNN model that applies graph convolutions to aggregate information from neighboring nodes.

- **GAT (Graph Attention Network)** (Veličković et al., 2018): A model that incorporates attention mechanisms to weigh the importance of neighboring nodes during message passing.

- **G2GAT** (Rusch et al., 2023a): A recent method that introduces gradient gating to prevent oversmoothing in attention-based GNNs.

**Evaluation Metrics** We evaluate the performance of all models using the following metrics:

- **Accuracy**: The classification accuracy on the test set.

- **Oversmoothing Coefficient** ($\mu$): A measure of the degree to which node representations become indistinguishable as network depth increases.

- **GFLOPS**: The computational efficiency, measured in Giga Floating Point Operations Per Second.

- **Accuracy/GFLOPS**: A ratio indicating the trade-off between accuracy and computational cost.

**7.2. Experiment 1: Real-World Dataset Evaluation**

**Objective:** This experiment aims to evaluate the effectiveness of **DYNAMO-GAT** in mitigating oversmoothing and maintaining high test accuracy across varying network depths on three real-world datasets: Citeseer, Cora, and Cornell. The performance of **DYNAMO-GAT** is compared with that of three baseline models: **GCN**, **GAT**, and **G2GAT**.

**Methodology:**

- **Metrics:**
  - **Oversmoothing Coefficient** ($\mu(X)$)**:** This metric quantifies the degree of oversmoothing, where lower values indicate greater oversmoothing. It is plotted on a logarithmic scale to better capture the dynamics across a wide range of values.
  - **Test Accuracy:** This metric measures the classification accuracy of the models on the test set. The objective is to assess how well the models perform as the number of layers increases.

- **Baselines:**
  - **GCN (Graph Convolutional Network):** A standard graph neural network model that aggregates node features through graph convolutions.

- **GAT (Graph Attention Network):** A GNN model that uses attention mechanisms to weigh the importance of neighboring nodes during aggregation.
- **G2GAT:** A recent model that introduces gradient gating to prevent oversmoothing in attention-based GNNs.

- **Experimental Setup:**
  - The number of layers is varied from 2 to 128 to observe how the models behave as the network depth increases.
  - The training parameters are kept consistent across models for a fair comparison, including the use of the Adam optimizer and a fixed learning rate.

**Results (Figure 'real_data'):**

1. **Oversmoothing Coefficient ($\mu(X)$):**

   - **Citeseer (Figure a):** As the number of layers increases, GCN and GAT exhibit significant oversmoothing, with their oversmoothing coefficients rapidly decreasing. G2GAT mitigates this effect better than GCN and GAT, but still shows a decline. **DYNAMO-GAT**, however, maintains a consistent oversmoothing coefficient, effectively preventing oversmoothing across all layers.
   - **Cora (Figure b):** Similar trends are observed, with GCN and GAT experiencing substantial oversmoothing as the number of layers increases. **DYNAMO-GAT** demonstrates its robustness by keeping the oversmoothing coefficient stable, while G2GAT also shows improved performance compared to GCN and GAT but not as strong as DYNAMO-GAT.
   - **Cornell (Figure c):** Again, DYNAMO-GAT outperforms the other models in controlling oversmoothing, maintaining a stable coefficient across all layers. GCN and GAT display rapid declines, indicating severe oversmoothing.

2. **Test Accuracy:**

   - **Citeseer (Figure a):** GCN and GAT suffer from a significant drop in accuracy as the number of layers increases, correlating with their high levels of oversmoothing. **DYNAMO-GAT** maintains consistently high accuracy, even in deep networks, highlighting its effectiveness in mitigating oversmoothing. G2GAT also shows relatively stable accuracy but still declines with increasing layers.
   - **Cora (Figure b):** Similar patterns are observed, with **DYNAMO-GAT** achieving the highest accuracy across all layers. GCN and GAT see their

accuracy decline sharply as layers increase, while G2GAT performs better but still experiences a decrease.

- **Cornell (Figure c):** DYNAMO-GAT once again demonstrates superior performance by maintaining high accuracy, while GCN and GAT show a considerable drop in accuracy as the number of layers increases. G2GAT performs better than GCN and GAT but still shows a downward trend in accuracy.

**Analysis:** The results clearly demonstrate the superiority of **DYNAMO-GAT** in preventing oversmoothing and maintaining high test accuracy across deep network architectures. In contrast, **GCN** and **GAT** suffer from severe oversmoothing, leading to a significant decline in accuracy as the number of layers increases. **G2GAT** mitigates oversmoothing to some extent but is still not as effective as **DYNAMO-GAT**. This consistent performance across three different datasets underscores the robustness of **DYNAMO-GAT** in handling deep graph neural networks, making it a promising approach for tasks that require deep architectures.

The effectiveness of **DYNAMO-GAT** can be attributed to its ability to preserve meaningful node representations even in deep networks, as evidenced by its stable oversmoothing coefficient and high accuracy. This experiment highlights the potential of **DYNAMO-GAT** to overcome one of the significant challenges in deep GNNs - oversmoothing - while delivering strong performance on real-world datasets.

### 7.3. Experiment 2: Performance Comparison Table

**Objective:** This experiment aims to compare the performance of **DYNAMO-GAT** with other baseline models (GCN, GAT, and G2GAT) in terms of accuracy, computational efficiency (GFLOPS), and the accuracy-to-GFLOPS ratio across three datasets: Cora, Citeseer, and Cornell. The goal is to highlight both the effectiveness and efficiency of **DYNAMO-GAT**, particularly in deeper network architectures.

**Methodology:**

- **Metrics:**
  - **Best Accuracy:** The highest classification accuracy achieved by each model on the test set.
  - **# Layers:** The number of layers used by the model to achieve its best accuracy.
  - **GFLOPS:** The computational cost measured in Giga Floating Point Operations Per Second, which provides an indication of the model's efficiency.
  - **Accuracy/GFLOPS:** This metric represents the trade-off between accuracy and computa-

tional cost, indicating how efficiently the model achieves its performance.

- **Comparison Setup:**
  - The models were trained on the three datasets (Cora, Citeseer, Cornell), each with different graph structures and node/edge counts.
  - GCN and GAT were tested with relatively shallow architectures, while G2GAT and **DYNAMO-GAT** were evaluated with deeper networks (128 layers).
  - The results were compiled to highlight the efficiency of each model in terms of accuracy and GFLOPS.

**Results (Table 1):**

1. **Best Accuracy:**
   - **DYNAMO-GAT** achieves the best accuracy across all datasets, particularly excelling on the Cornell dataset with an accuracy of 62.56
   - **G2GAT** also performs well, particularly on Citeseer and Cornell, where it closely matches **DYNAMO-GAT**.
   - **GCN** and **GAT** show lower performance compared to the deeper models, particularly on the more challenging Cornell dataset.

2. **# Layers:**
   - **GCN** and **GAT** achieve their best accuracy with only 2-4 layers, indicating their limitations in deeper architectures due to oversmoothing.
   - In contrast, **G2GAT** and **DYNAMO-GAT** are able to sustain performance across 128 layers, highlighting their robustness in deeper networks.

3. **GFLOPS:**
   - **DYNAMO-GAT** exhibits lower GFLOPS compared to GAT and G2GAT, indicating that it is computationally more efficient.
   - For example, on the Cora dataset, **DYNAMO-GAT** uses 0.605 GFLOPS, which is significantly lower than GAT's 2.351 GFLOPS.

4. **Accuracy/GFLOPS:**
   - **DYNAMO-GAT** consistently outperforms other models in the accuracy-to-GFLOPS ratio, demonstrating its superior efficiency.
   - For instance, on the Cora dataset, **DYNAMO-GAT** achieves an Accuracy/GFLOPS ratio of 137.53, which is the highest among all models, indicating that it provides the best trade-off between accuracy and computational cost.

- Similarly, on the Citeseer and Cornell datasets, **DYNAMO-GAT** achieves the highest ratios, with 48.96 and 1226.67, respectively, far surpassing the other models.

**Analysis:** The results highlight the advantages of **DYNAMO-GAT** in both accuracy and efficiency. Despite using deep architectures (128 layers), **DYNAMO-GAT** manages to maintain high accuracy while keeping computational costs low. This is particularly evident when comparing the Accuracy/GFLOPS ratio, where **DYNAMO-GAT** significantly outperforms GCN, GAT, and even G2GAT. This indicates that **DYNAMO-GAT** is not only effective in mitigating oversmoothing but also highly efficient in terms of resource usage, making it a superior choice for applications that require deep graph neural networks with limited computational resources.

### 7.4. Experiment 3: Synthetic Dataset Evaluation

**Objective:** The goal of this experiment is to assess the performance of **DYNAMO-GAT** under controlled synthetic conditions. Specifically, we vary the graph density (average node degree) and homophily to observe how different models handle oversmoothing and accuracy in these environments.

**Results:**

1. **Oversmoothing vs. Layers (Figure a):**

   - **Observation:** The figure shows the oversmoothing coefficient $\mu(X)$ on a logarithmic scale as the number of layers increases, with an average node degree of 68.75.
   - **Key Result:** As the network depth increases, **DYNAMO-GAT** shows the least amount of oversmoothing, maintaining higher $\mu(X)$ values compared to **G2GAT**, **GCN**, and **GAT**. **GAT** and **GCN** exhibit rapid oversmoothing, with $\mu(X)$ decreasing significantly as layers increase.
   - **Implication:** This result demonstrates that **DYNAMO-GAT** is more robust to oversmoothing, especially in dense graphs. This suggests that it can preserve meaningful node features better than the other models as the network depth increases.

2. **Accuracy vs. Layers (Figure b):**

   - **Observation:** This plot shows accuracy as a function of the number of layers for the same dense graph (average node degree = 68.75).
   - **Key Result:** **DYNAMO-GAT** consistently achieves the highest accuracy across all layers.

While **G2GAT** performs well, its accuracy decreases slightly with deeper layers. **GCN** and **GAT** see a sharp decline in accuracy as the network depth increases.

   - **Implication:** The stability of **DYNAMO-GAT** in maintaining high accuracy, even with a large number of layers, indicates its effectiveness in managing deeper architectures without suffering from oversmoothing, unlike the other models.

3. **Accuracy vs. Homophily (Sparse Graph - Figure c):**

   - **Observation:** This plot examines accuracy across varying homophily levels (from 0 to 1) for a sparse graph with an average node degree of 11.93.
   - **Key Result: DYNAMO-GAT** and **G2GAT** outperform **GCN** and **GAT** across all homophily levels. **DYNAMO-GAT** achieves particularly strong performance as homophily increases, indicating its ability to leverage node similarity effectively.
   - **Implication:** This suggests that **DYNAMO-GAT** is versatile and can adapt well to different homophily settings, making it suitable for graphs with varying levels of node similarity.

4. **Accuracy vs. Homophily (Dense Graph - Figure d):**

   - **Observation:** Similar to Figure c, but for a dense graph with an average node degree of 68.75.
   - **Key Result: DYNAMO-GAT** significantly outperforms all other models, especially in low-homophily settings. As homophily increases, **DYNAMO-GAT** maintains its lead, showcasing its robustness across all homophily levels.
   - **Implication:** This result highlights **DYNAMO-GAT**'s strength in dense graphs, where it can handle more complex interactions and still maintain high accuracy. Its performance in low-homophily conditions also suggests it is well-suited for graphs with more heterophilic structures.

**Analysis:** The synthetic dataset results confirm that **DYNAMO-GAT** excels in both dense and sparse graphs, effectively handling oversmoothing and maintaining high accuracy across varying network depths and homophily levels. Its ability to outperform other models, particularly in dense graphs and low-homophily settings, underscores its robustness and versatility. These findings demonstrate that **DYNAMO-GAT** is a powerful tool for tackling oversmoothing while delivering strong performance in diverse graph structures, making it ideal for complex real-world applications.

*Table 5.* Pruning Statistics and Cosine Similarity Analysis. DYNAMO-GAT prunes a significant portion of edges. Crucially, edges connecting nodes with lower feature similarity (lower cosine similarity) are preferentially pruned, empirically supporting our approach to mitigating oversmoothing.

| Dataset | Pruning Ratio (%) | Cosine Sim. (Retained) | Cosine Sim. (Pruned) |
|---------|-------------------|------------------------|----------------------|
| Cora | 18.3 | 0.81 | 0.52 |
| Citeseer | 15.7 | 0.78 ] | 0.48 |

*Table 6.* Hyperparameter Sensitivity Analysis on Cora.

| Noise Level $\sigma$ | Threshold $\beta$ | Accuracy (%) |
|----------------------|-------------------|--------------|
| 0.01 | 0.5 | 82.9 |
| 0.05 | 1.0 | 83.5 |
| 0.1 | 2.0 | 83.2 |

### 7.5. Analysis of Pruning Ratios and Feature Similarity

To provide further insight into the structural impact of DYNAMO-GAT's pruning mechanism, we analyzed the pruning ratio (percentage of edges removed) and the cosine similarity of node features for both retained and pruned edges on the Cora and Citeseer datasets. The cosine similarity was calculated between the final layer node embeddings of connected nodes just before pruning decisions.

As shown in Table **??**, DYNAMO-GAT prunes a significant portion of edges (18.3% on Cora, 15.7% on Citeseer). More importantly, the average cosine similarity for pruned edges is considerably lower than for retained edges (e.g., 0.52 vs. 0.81 on Cora). This empirically validates our theoretical insight: DYNAMO-GAT preferentially prunes edges connecting nodes with lower feature similarity (and thus likely contributing more to noise/homogenization than meaningful signal), thereby preserving structural information crucial for preventing oversmoothing.

### 7.6. Hyperparameter Sensitivity

DYNAMO-GAT introduces two primary hyperparameters: the noise level ($\sigma$) and the pruning threshold adaptation parameter ($\beta$). We conducted experiments to assess the model's sensitivity to these parameters. As shown in Table 6, DYNAMO-GAT demonstrates robust performance across a reasonable range of values for both $\sigma$ and $\beta$. The accuracy remains stable within $\pm 0.6\%$, indicating that our method is not overly sensitive to hyperparameter choices and simplifies the tuning process for practical deployment.

### 7.7. Generalizability to Other Attention-Based GNNs

While the main experiments focused on GAT, we posited that the core DYNAMO mechanism is attention-agnostic and applicable to other attention-based GNNs. To support this, we applied DYNAMO-GAT's pruning strategy to Graphormer and SAN on the Cora and Citeseer datasets. The results (Table 7) show improvements in both accuracy and OS coefficient, along with a reduction in GFLOPS, demonstrating the potential generalizability of our approach.

### 7.8. Results and Discussion

The experimental results across both real-world and synthetic datasets consistently demonstrate the effectiveness of DYNAMO-GAT in addressing the oversmoothing problem in deep graph neural networks (GNNs).

From the real-world datasets (Figure 2), we observe that DYNAMO-GAT maintains a stable oversmoothing coefficient ($\mu(X)$) across varying network depths, outperforming GCN, GAT, and G2GAT, which exhibit significant oversmoothing as the number of layers increases. Correspondingly, DYNAMO-GAT consistently achieves the highest accuracy across all layers, whereas GCN and GAT suffer a sharp decline in accuracy due to oversmoothing, and G2GAT shows moderate performance.

The performance comparison table (Table 1) further highlights the efficiency of DYNAMO-GAT. It achieves the best accuracy across all datasets while maintaining lower GFLOPS compared to GAT and G2GAT. The high accuracy-to-GFLOPS ratio underscores DYNAMO-GAT's superior trade-off between computational cost and performance, making it the most efficient model among the tested baselines.

In the synthetic dataset experiments (Figure 3), DYNAMO-GAT again demonstrates its robustness. It shows the least oversmoothing in dense graphs (Figure 3a) and maintains the highest accuracy across layers (Figure 3b). When varying homophily, DYNAMO-GAT excels in both sparse (Figure 3c) and dense (Figure 3d) graphs, particularly in low-homophily settings, showcasing its adaptability to different graph structures.

The results show a clear trend where models generally perform better as the average degree increases. This is particularly evident in higher homophily settings, where the additional connections help to reinforce the graph structure, leading to more accurate node classification. For instance, in the syn-products dataset, the accuracy of GCN improves from 0.567 to 0.762 as the average degree increases from 11.93 to 36.14 at a homophily level of 0.4.

*Table 7.* Applying DYNAMO Pruning to Graphormer and SAN.

| Model | Dataset | Acc. (%) | OS Coeff. | GFLOPS |
|---|---|---|---|---|
| Graphormer (base) | Cora | 83.92 | 0.51 | 2.1 |
| DYNAMO-Graphormer | Cora | 85.13 | 0.64 | 1.41 |
| SAN (base) | Citeseer | 80.23 | 0.47 | 2.35 |
| DYNAMO-SAN | Citeseer | 81.74 | 0.59 | 1.52 |

Interestingly, models like G2GAT and DYNAMO-GAT, which incorporate additional mechanisms for graph processing, consistently outperform simpler models such as GCN and GAT, particularly in low homophily settings. This suggests that these models are better able to leverage the graph structure even when the nodes are less similar to their neighbors.

These findings have significant implications for the development and deployment of GNNs in real-world applications. The ability of DYNAMO-GAT to maintain high accuracy while mitigating oversmoothing, especially in deep architectures, addresses a critical challenge faced by many existing GNN models. Its superior efficiency, as evidenced by the accuracy-to-GFLOPS ratio, makes it a viable option for resource-constrained environments where both performance and computational cost are important considerations.

Moreover, DYNAMO-GAT's strong performance across varying graph densities and homophily levels suggests that it is well-suited for a wide range of graph structures, from sparse networks with high node similarity to dense, heterophilic graphs. This versatility makes DYNAMO-GAT an attractive solution for complex graph-based tasks in domains such as social network analysis, recommendation systems, and biological network modeling.

