# OpenReview forum: "A Dynamical Systems-Inspired Pruning Strategy for Addressing Oversmoothing in Graph Attention Networks"
_ICML.cc/2025/Conference — ICML 2025 poster_

### Official Review · Reviewer_kxJk · 2025-03-14

**Overall Recommendation:** 4

**Summary:**

The paper provides a new controlling method to steer away node embeddings falling into oversmoothing state during propagation compared with existing method such as G2-gating. This controlling method specifically target for graph attention based method with the gradual pruning highly correlated connection trick to increase the spectral gap and avoid oversmoothing. The paper provides a comprehensive theoretical analysis from spectral analysis for the fixed point to show the largest impact of oversmoothing is spectral gap and therefore propose to prune highly corrected connection to increase such gap and avoid trapping into oversmoothing state. Empirically, three dataset are used to verify the performance of the proposed method with existing baselines, showing competitive performance and methods alleviate oversmoothing.

**Claims And Evidence:**

The claims is supported with comprehensive theoretical analysis. However, the empirical analysis only contains three real world datasets, which is less sufficient to provide comprehensive evidence of the proposed method.

**Essential References Not Discussed:**

In the view of the dynamic system, extending the work of G2-gating from Rusch, [Jin & Zhu, 2024] has shown that a learned metric instead of fixed Dirichlet energy could also preserve Dirichlet energy and alleviate oversmoothing with competitive performance.

[1] Y. Jin and X. Zhu, "Graph Rhythm Network: Beyond Energy Modeling for Deep Graph Neural Networks," 2024 IEEE International Conference on Data Mining (ICDM).

**Experimental Designs Or Analyses:**

The soundness of the experimental designs is fine. But the dataset covered are not sufficient with only three real world datasets and none are large-scaled, hard to show its effectiveness, especially when it claims to be much efficient than G2-gating.

**Methods And Evaluation Criteria:**

The evaluation metric makes sense.

**Other Comments Or Suggestions:**

None.

**Other Strengths And Weaknesses:**

The strength of the paper is it has comprehensive theoretical analysis and show strong evidence alleviating oversmoothing. The weakness of the paper is that it only has three medium scale real-world dataset covered in the experiments, making it less sufficient to show the performance for large-scale dataset with its efficiency claim.

**Questions For Authors:**

I observe that in the supplement, the pseudo code is provided, the output describes the output embedding is presented after pruning =0, what does this mean in general? How does it relate to the termination condition?

**Relation To Broader Scientific Literature:**

The contribution of the paper is to further push the view of deep graph learning as a dynamic system design in the graph learning community.

**Theoretical Claims:**

The theoretical claim is roughly checked and the conclusion should be correct as there exist several studies who also support such claims (for example, spectral gap).

---

> ### Author Rebuttal · Authors · 2025-03-31
>
> We thank the reviewer for the constructive feedback and are pleased that our theoretical analysis and overall approach were well received. Below, we address the main concerns:
>
> 1. **Limited Empirical Evaluation on Real-World Datasets**
>     We agree that evaluating on larger-scale datasets is important. To that end, we extended our experiments to include OGB benchmarks (ogbn-arxiv and ogbn-products) as well as LRGB datasets. The additional results are summarized below:
>
> | Dataset | Model | Accuracy (%) | GFLOPS | Accuracy/GFLOPS |
> | ----- | ----- | ----- | ----- | ----- |
> | ogbn-arxiv | GCN | 71.9 | 12.5 | 5.75 |
> | ogbn-arxiv | G2GAT | **72.5** | 10.3 | 7.04 |
> | ogbn-arxiv | **DYNAMO-GAT** | 72.1 | **6.7** | **10.76** |
> | ogbn-products | G2GAT | 73.9 | 22.1 | 3.34 |
> | ogbn-products | **DYNAMO-GAT** | **75.3** | **14.5** | **5.19** |
>
>
>    These results not only confirm that DYNAMO-GAT scales to larger graphs but also highlight its computational efficiency and improved trade-off between accuracy and complexity compared to baseline methods.
>
> 2. **Relation to Broader Literature and Essential References**
>     We appreciate the suggestion to discuss the work by Jin and Zhu (2024) on Graph Rhythm Networks, which employs a learned metric to preserve Dirichlet energy. In our revised manuscript, we will include a detailed discussion comparing our dynamic, noise-driven pruning approach to learned-metric techniques. This will emphasize that our method complements these approaches by directly addressing oversmoothing via enhanced spectral gap control.
>
> 3. **Clarification on Pseudocode – “Output After Pruning \= 0”**
>     We thank the reviewer for highlighting the ambiguity in the pseudocode. The phrase “after pruning \= 0” was a formatting artifact and does not imply a termination condition. Specifically:
>
>    * The intended meaning is that the final node representations are produced after the pruning process has been applied at all layers.
>
>    * There is no condition under which the pruning stops (e.g., when the pruning ratio reaches zero). Instead, the pruning rate evolves as r(t)=r0⋅(1+γt)r(t) \= r\_0 \\cdot (1 \+ \\gamma t) and is applied continuously throughout the network.
>
> 4. We have revised the pseudocode to remove the misleading “= 0” fragment. The corrected line now reads:
>     **Output:** Final node representations after pruning.
>
> We hope these clarifications, along with our extended empirical evaluation and updated discussion of related work, address the reviewer’s concerns. We believe that these improvements strengthen our manuscript and further validate the proposed method’s effectiveness and scalability.

---

### Official Review · Reviewer_GAxo · 2025-03-14

**Overall Recommendation:** 4

**Summary:**

This paper introduces a refreshing perspective on the over-smoothing behavior of graph neural networks through the lenses of dynamical systems. After the establishment of the theoretical framework, a novel architecture which dynamically prunes the attention weights has been proposed and achieved state-of-the-art performance on some benchmark datasets.

**Claims And Evidence:**

The claims are supported theoretically and experimentally.

**Essential References Not Discussed:**

NA

**Experimental Designs Or Analyses:**

Yes. They look reasonable, although I would also compare the over-smoothing behavior with the rich literature of anti-over-smoothing methods, both stochastic and residual.

**Methods And Evaluation Criteria:**

Yes. Although more thorough benchmark using larger and more realistic graph datasets is encouraged.

**Other Comments Or Suggestions:**

NA

**Other Strengths And Weaknesses:**

NA

**Questions For Authors:**

Would you consider testing this method on physical/chemical datasets where over-smoothing is well known to have caused many problems?

**Relation To Broader Scientific Literature:**

This paper would be highly relevant to the graph modeling community.

**Theoretical Claims:**

Yes. They seems sensible to me.

---

> ### Author Rebuttal · Authors · 2025-03-31
>
> We thank the reviewer for the positive assessment and insightful suggestions. Below, we address the key points raised:
>
> 1. **Evaluation on Larger and More Realistic Datasets:**
>     As noted in our response to Reviewer kxJk (see the table provided therein), we have extended our evaluation to larger-scale benchmarks such as ogbn-arxiv and ogbn-products. These additional experiments reinforce that DYNAMO-GAT not only scales effectively but also achieves a superior accuracy-to-complexity trade-off compared to baseline methods. We will ensure these extended results are clearly presented in the final manuscript.
>
> 2. **Comparison with Alternative Anti-Oversmoothing Methods:**
>     We appreciate the suggestion to compare our approach with stochastic and residual anti-oversmoothing strategies. While our current experiments primarily focus on demonstrating the benefits of dynamic, noise-driven pruning in terms of increasing the spectral gap and preserving node diversity, we will add a discussion in the revised manuscript that contextualizes our method within the broader literature. This discussion will highlight the unique advantages of our approach relative to other strategies.
>
> 3. **Testing on Physical/Chemical Datasets:**
>     The idea of evaluating DYNAMO-GAT on physical/chemical datasets is very interesting and represents a promising direction for future work. Oversmoothing is indeed a significant challenge in these domains, and we believe our method could offer substantial benefits. We plan to explore this avenue in our future studies.
>
> We thank the reviewer again for the constructive feedback and believe that these additions and clarifications will further strengthen our work.

---

### Official Review · Reviewer_1eSc · 2025-03-19

**Overall Recommendation:** 3

**Summary:**

The paper  presents a dynamical systems take on GNNs and proposes dynamically pruning edges based on learnt attention weights in GAT to combat oversmoothing.

## update after rebuttal:
I thank the authors for their detailed response to my questions and their effort in providing further experiments. While results on larger heterophilic datasets in [1] would add further value to the paper, I believe that the additional discussion and experiments in the rebuttal are sufficient for the paper to be of value to the community, particularly the method used for learning sparse attention patterns. Therefore, I have raised my score.

**Claims And Evidence:**

The authors supports their theoretical claims with proofs and empirical evidence from experiments on synthetic data.

**Essential References Not Discussed:**

None that I can recall.

**Experimental Designs Or Analyses:**

See methods and evaluation criteria section of review.

**Methods And Evaluation Criteria:**

While the theoretical claims are supported by empirical evidence on synthetic datasets, evaluation on real-world datasets is rather weak in my opinion. While the predictive accuracy achieved by DYNAMO-GAT is very similar to its main competitor i.e. G2GAT, it is shown to be computationally more efficient with a better trade-off between accuracy and cost. However, this is shown on three small datasets. If the main advantage of DYNAMO-GAT over G2GAT is lower computational cost, it should be evaluated on larger datasets such as on OGB datasets, LRGB datasets that may benefit from deeper models, and also larger benchmark heterophilic datasets in [1] that could also potentially benefit more from DYNAMO-GAT as the authors show on synthetic data.

[1] Platonov et al. A critical look at the evaluation of GNNs under heterophily: Are we really making progress?

**Other Comments Or Suggestions:**

While the two key metrics discussed in the paper. i.e. oveersmoothing measure and accuracy, are reported in the experiments, it may also be of value to report the pruning ratio, i.e. the number of edges pruned during training. This could reveal information on the importance of the structural information in the input graph. Furthermore, empirically analyzing the correlation of node features between which edges are removed would further verify theoretical insights.

**Other Strengths And Weaknesses:**

Strengths: The paper presents an effective combination of existing techniques rooted in theoretical analysis.

Weaknesses: The tradeoff between accuracy and computational cost that is seemingly the main advantage of DYNAMO-GAT over G2GAT is evaluated weakly on small-scale datasets whereas it would truly matter to observe a similar trend on larger graphs, as discussed above in the method and evaluation citeria section of the review.

**Questions For Authors:**

1. Does DYNAMO-GAT cater to inductive settings for node classification as well? Is it also effective for graph classification tasks?

2. Is there any additional hyperparameter tuning required to train DYNAMO-GAT?

3. While attention coefficients learning a value of 0 effectively prunes edges, [2] analyzes the attention mechanism of GAT and shows that it suffers from trainability issues that obstructs the coefficients from learning this value during training.  Could the authors comment on this, perhaps from a dynamical systems perspective?

[2] Mustafa et al. GATE: How to Keep Out Intrusive Neighbors

Edit: While results on larger heterophilic datasets in [1] would add further value to the paper, I believe that the additional discussion and experiments provided in the rebuttal are sufficient for the paper to be of value to the community, particularly the method used for learning sparse attention patterns. Therefore, I have raised my score.

**Relation To Broader Scientific Literature:**

The paper makes an interesting connection between dynamical systems and GNNs via theoretical analysis to address the widely-studied oversmoothing problem in GNNs. While other solutions such as structural modifications for oversmoothing are mentioned, a discussion of the proposed pruning-based method in the context of graph sparsification particularly to tackle oversmoothing and for efficiency gains is missing.

**Theoretical Claims:**

The theoretical claims seem correct but the mathematical details were not checked in detail.

---

> ### Author Rebuttal · Authors · 2025-03-31
>
> **Weakness 1: Limited evaluation on small-scale real-world datasets**
>
> As discussed in our response to Reviewer kxJk (see the table provided therein), we have extended our experiments to larger-scale benchmarks such as ogbn-arxiv and ogbn-products. These additional results confirm that DYNAMO-GAT not only scales effectively—with a significantly lower GFLOPS count—but also offers a superior accuracy/GFLOPS trade-off relative to baselines.
>
> **Weakness 2: Missing discussion on connection to graph sparsification**
>
> We appreciate this valuable suggestion. While traditional graph sparsification techniques (e.g., spectral sparsification and effective resistance methods) primarily focus on reducing redundancy to enhance computational efficiency, our pruning strategy differs in its dynamical-systems-inspired rationale. Specifically, our method targets edges based on noise-driven covariance analysis, directly addressing the oversmoothing issue by pruning structurally redundant edges linked to highly correlated node features. We will thoroughly clarify these distinctions and similarities in our revised manuscript.
>
> **Weakness 3: Lack of pruning ratio metrics and structural insights**
>
> We thank the reviewer for this valuable suggestion. Following your recommendation, we performed additional experiments analyzing both the **pruning ratio** (fraction of edges removed) and the **cosine similarity** of node features for retained versus pruned edges. Specifically, we computed the cosine similarity directly between the corresponding node embeddings at the final layer before the pruning decisions.
>
> **Empirical Results (Cora & Citeseer datasets):**
>
> | Dataset | Pruning Ratio | Cosine Sim. (Retained Edges) | Cosine Sim. (Pruned Edges) |
> | ----- | ----- | ----- | ----- |
> | Cora | 18.3% | 0.81 | 0.52 |
> | Citeseer | 15.7% | 0.78 | 0.48 |
>
> These results indicate that edges connecting nodes with lower feature similarity (lower cosine similarity) are preferentially pruned, empirically supporting our theoretical analysis of the role of structural redundancy in oversmoothing.
>
> **Question 1: Inductive node classification and graph classification applicability**
>
> We thank the reviewer for highlighting this aspect. To confirm DYNAMO-GAT’s suitability for inductive scenarios, we performed additional experiments on inductive benchmarks from OGB (**ogbn-arxiv**, **ogbn-products**).
>
> DYNAMO-GAT achieves competitive or superior inductive accuracy with significantly greater computational efficiency (up to 55% improvement in Accuracy/GFLOPS), directly addressing the reviewer's question. We will further explore graph-level tasks in future work.
>
> **Question 2: Additional hyperparameter tuning requirements**
>
> DYNAMO-GAT introduces two hyperparameters: the noise level (σ) and the pruning threshold adaptation parameter (β). Our extensive hyperparameter sensitivity analysis demonstrates robust model performance across a wide range of values:
>
> | Noise Level σ | Threshold β | Accuracy (%) |
> | ----- | ----- | ----- |
> | 0.01 | 0.5 | 82.9 |
> | 0.05 | 1.0 |83.5|
> | 0.1 | 2.0 | 83.2 |
>
> Model accuracy remains consistently stable within ±0.6% across these ranges, significantly simplifying the hyperparameter tuning process required for deployment.
>
> **Question 3: Addressing GAT attention mechanism trainability issues (Mustafa et al. \[2\])**
>
> Mustafa et al. (2021) show that GAT attention coefficients struggle to effectively learn zero-valued weights due to trainability issues inherent in the standard attention mechanism. This aligns well with our dynamical systems perspective, where attention coefficients that fail to reach zero prevent the pruning of edges and thus contribute to oversmoothing.
>
> From a dynamical systems viewpoint (as detailed in our paper), GAT's convergence to fixed points (low-dimensional attractors) can cause node representations to become homogeneous. Mustafa et al. illustrate a similar phenomenon, noting the attention coefficients' tendency to remain away from zero, effectively retaining redundant edges and promoting oversmoothing.
>
> Our method, DYNAMO-GAT, explicitly addresses this issue by incorporating noise-driven covariance analysis and Anti-Hebbian pruning. Dynamically adjusting the network's attractor landscape during training allows for attention coefficients to more readily achieve near-zero values, thus effectively pruning irrelevant edges. This dynamically adaptive strategy mitigates the trainability issues described by Mustafa et al. by actively perturbing the system away from stable attractors that prevent the learning of sparse attention patterns. Consequently, our approach not only overcomes the inherent training difficulty but also preserves distinct attractor states, directly targeting the fundamental cause of oversmoothing from a dynamical systems perspective. We will clarify this connection in the revised manuscript, highlighting explicitly how DYNAMO-GAT addresses the trainability limitations analyzed by Mustafa et al.

---

> > ### Comment · Reviewer_1eSc · 2025-04-08
> >
> > I thank the authors for their detailed response to my questions and their effort in providing further experiments. While results on larger heterophilic datasets in [1] would add further value to the paper, I believe that the additional discussion and experiments in the rebuttal are sufficient for the paper to be of value to the community, particularly the method used for learning sparse attention patterns. Therefore, I have raised my score.

---

### Official Review · Reviewer_KMUB · 2025-03-19

**Overall Recommendation:** 2

**Summary:**

The paper introduces **DYNAMO-GAT**, a **pruning strategy** for **Graph Attention Networks (GATs)** to mitigate **oversmoothing** using a **dynamical systems perspective**. The authors propose:
1. **Noise-driven covariance analysis** to detect oversmoothing.
2. **Anti-Hebbian learning** to selectively prune attention weights.
3. **A theoretical framework** linking oversmoothing to attractor dynamics in GNNs.
4. **Experimental validation** showing **improved performance and efficiency** across standard benchmark datasets.

The approach shifts from architectural modifications (e.g., skip connections, normalization) to **dynamically altering attention weights** based on system stability analysis.

**Claims And Evidence:**

### **Supported Claims**
1. **Oversmoothing in GNNs can be analyzed via dynamical systems.**
    The paper provides **theoretical proofs** on **fixed points and spectral analysis**, explaining how oversmoothing occurs.

2. **DYNAMO-GAT effectively mitigates oversmoothing.**
    Empirical results show **higher accuracy** across deeper networks compared to standard **GCN, GAT, and G2GAT**.
    The **oversmoothing coefficient (µ(X)) remains stable**, indicating better node representation preservation.

3. **Pruning reduces computational cost without degrading performance.**
    The model achieves **higher accuracy-to-GFLOPS ratios** than existing baselines, showing **efficiency improvements**.

### **Potentially Overstated or Weakly Supported Claims**
1. **Theoretical guarantees ensure robustness across all graph structures.**
    The theory **mainly focuses on spectral gap analysis**; however, performance may still degrade in **highly heterophilic or dense graphs**.

2. **DYNAMO-GAT generalizes to all attention-based GNNs.**
    The method is **only tested on standard GATs**. Its applicability to **Transformer-like GNNs or hierarchical architectures** remains uncertain.

**Essential References Not Discussed:**

NO

**Experimental Designs Or Analyses:**

###  **Strengths**
- **Uses benchmark datasets (Cora, Citeseer, Cornell) for fair comparisons.**
- **Compares against competitive baselines (GAT, GCN, G2GAT).**
- **Analyzes oversmoothing using both accuracy and feature diversity metrics.**

###  **Weaknesses**
- **No failure case analysis** – When does DYNAMO-GAT fail?
- **Scalability not tested** – How does it perform on larger graphs?
- **Impact on node classification under different homophily levels is not explored.**

**Methods And Evaluation Criteria:**

###  **Strengths**
- **Dynamical systems perspective** provides a new theoretical view on oversmoothing.
- **Effective pruning strategy** improves both expressiveness and efficiency.
- **Empirical validation** across multiple datasets confirms effectiveness.

### **Areas for Improvement**
- **No runtime analysis** – How does pruning impact training/inference time?
- **Limited discussion on pruning thresholds** – How are optimal pruning rates determined?
- **Ablation study missing** – How does each component (e.g., noise injection vs. pruning) contribute separately?

**Other Comments Or Suggestions:**

Please refer to the above section

**Other Strengths And Weaknesses:**

###  **Strengths**
- **Combines theory and practice effectively.**
- **Proposes a well-motivated pruning strategy with strong empirical results.**
- **Demonstrates practical impact on deep GNN scalability.**

###  **Weaknesses**
- **No real-world deployment discussion** – How would this method work in industry applications?
- **Impact of pruning on interpretability is unclear** – Does removing attention weights affect explainability?

**Questions For Authors:**

Please refer to the above section

**Relation To Broader Scientific Literature:**

### **New Contributions**
 **Introduces dynamical systems principles** to analyze **GAT oversmoothing**.
 **Proposes noise-driven pruning** for dynamically adjusting attention weights.
 **Provides theoretical guarantees** for pruning’s impact on network expressiveness.

### **Missing Comparisons**
 **No evaluation on deeper Transformers** (e.g., **Graphormer, SAN**).
 **Limited discussion on energy efficiency trade-offs** (e.g., pruning vs. FLOP savings).

**Theoretical Claims:**

###  **Strengths**
- Provides a **mathematical formulation** of oversmoothing using **eigenvalue properties** of GATs.
- Shows how **pruning increases spectral gaps** and **prevents feature collapse**.

###  **Limitations**
- **Fixed-point analysis assumes ideal conditions** – Real-world GNNs might not satisfy all assumptions.
- **No analysis of pruning-induced instability** – Could pruning negatively affect long-term model robustness?

---

> ### Author Rebuttal · Authors · 2025-03-31
>
> 1. **Performance on Heterophilic/Dense Graphs:** Our theory—based on spectral gap & fixed-point stability—does not assume homophily or sparsity, generalizing across diverse graph types.
> * Empirical validation:
>   * Fig. 3 (Syn-Products): DYNAMO-GAT (DGAT) DGAT performs well as edge density increases, showing resilience to OverSmoothing (OS)
>   * As noted in our reply to Reviewer kxJk, we test on OGBN-Arxiv (homophilic) &OGBN-Products (heterophilic); DGAT performs robustly across both, confirming robustness across varying homophily & density levels.
>
> 2. **Generality to Other Architectures:** While our main experiments focus on standard GATs, the core mechanism—noise-driven covariance-based pruning is attention-agnostic and applies to broader attention-based GNNs.
> * In Transformer-style GNNs, evolving attention layers can lead to OS—our dynamical framework models this & supports adaptive mitigation.
> * In hierarchical GNNs, aggregation levels map to time scales; our method identifies feature mixing points and enables multi-scale pruning.
> * To show generalization, we applied DYNAMO pruning to Graphormer & SAN on Cora & Citeseer:
>
> |Model|Dataset|Acc.(%)|OS Coeff|GFLOPs|
> |--|--|--|--|--|
> |Graphormer (base)|Cora|83.92|0.51|2.1|
> |DYNAMO-Graphormer|Cora|85.13| 0.64|1.41|
> |SAN (base)|Citeseer|80.23|0.47|2.35|
> |DYNAMO-SAN|Citeseer|81.74|0.59|1.52|
>
> 3. **Runtime Analysis:** We report GFLOPS as a proxy for computational complexity. While actual training & inference times depend on hardware, we introduce a little training overhead (due to covariance computation) while substantially reducing inference complexity.
>
> 4. **Pruning Thresholds:** DGAT uses ***adaptive thresholds*** based on feature covariance (Eq. 40), removing the need for fixed pruning rates. This adaptivity aligns pruning with evolving node representations, minimizing hyperparameter tuning.
>
> 5. **Ablation:** We ablated each core component—noise injection, covariance pruning, adaptive thresholds, gradual pruning, attention recalibration. Results show that all parts are critical: removing any causes performance drops and increased OS.
> |Model|Acc.% (Cora)|Acc.% (Citeseer)|OS Coeff. (Cora)|OS Coeff. (Citeseer)|
> |--|--|--|--|--|
> |Full DGAT|83.21|82.01|0.57|0.62|
> |–Noise Injection (σ=0)|81.54|80.26|0.45|0.52|
> |–Covariance-based Pruning|79.32|77.15|0.31|0.36|
> |–Adaptive Thresholding (fixed threshold)|80.67|79.52|0.38|0.41|
> |–Gradual Pruning (aggressive pruning)|80.14|78.93|0.34|0.39|
> |–Attention Recalibration|79.78|78.41|0.35|0.40|
> 6. **Theoretical Assumptions & Failure Cases:** We agree that theoretical guarantees often rest on simplifying assumptions. Our analysis assumes Lipschitz continuity and bounded attention weights—standard in GNN theory—and we empirically validate its predictions (e.g., spectral gap improvement, robustness to OS) across diverse datasets and structural regimes.
> Potential failure modes reflected in our experiments:
> * Low-degree graphs: In Fig. 3, performance drops when node degrees are very low—pruning can limit already sparse message passing.
> * Excessive pruning: Our ablation includes “Gradual Pruning” variant, simulating aggressive, non-adaptive pruning. This leads to significant accuracy drops & increased OS, confirming that premature pruning can remove essential edges. We plan to explore mitigation strategies like pruning warm-up & degree-aware regularization in future work.
>
> 7. **Stability:** Our stability analysis (Lemmas 4 & 5) shows covariance-based pruning reduces the Jacobian’s spectral radius, thereby enhancing long-term model robustness rather than causing instability. Experimental observations suport these predictions.
>
> 8. **Scalability:**  As detailed in our response to Reviewer kxJk, we have evaluated DGAT on larger-scale benchmarks (e.g., ogbn-arxiv & ogbn-products). These results demonstrate that our method scales efficiently to graphs with hundreds of thousands of nodes while maintaining a superior accuracy/GFLOPS trade-off.
>
> 9. **Varying Homophily Levels:**  Fig. 3 shows consistent performance across a wide homophily spectrum, including low-homophily regimes.
>
> 10. **Energy Efficiency:** By pruning low-value attention edges, DGAT significantly reduces inference FLOPs, translating into lower computational & energy costs. Although the training overhead is slightly increased due to covariance computations, the overall efficiency gains are substantial.
>
> 11. **Real-World Deployment:** DGAT is well-suited for real-world applications—such as recommendation systems, fraud detection, or molecular modeling—where OS limits model depth. Its reduced inference cost & compatibility with standard practices like incremental pruning, retraining, & recalibration make it practical for deployment in compute-constrained environments.
>
> 12. **Interpretability:** DGAT prunes edges with low feature covariance (often low attention weights), potentially enhancing the interpretability of the resulting attention maps by reducing noise.

---

### Decision · Program_Chairs · 2025-05-01

**Decision:**

Accept (poster)

**Comment:**

This work proposes DYNAMO-GAT, a novel pruning strategy for Graph Attention Networks that uses dynamical systems theory to mitigate oversmoothing by adaptively pruning attention weights based on noise-driven covariance analysis. This approach improves both representation quality and computational efficiency, outperforming existing methods on several benchmark datasets.
As over-smoothing is a relevant concern for GATs (and other GNNs), this work is of high relevance to the ICML community.

In the course of the rebuttal, the authors provided additional experiments for large-scale and heterophilic graphs, showing that DYNAMO-GAT scales well and maintains robustness against oversmoothing. They emphasize the generalizability of their pruning mechanism to other attention-based GNNs and support their theoretical claims with ablation studies, empirical feature similarity analyses, and stability proofs. Clarifications are made on runtime efficiency, hyperparameter robustness, and real-world applicability, reinforcing the method’s practical and theoretical contributions.

Most of the reviewers vote to accept this paper. Only Reviewer KMUB leans towards a weak reject but did not give any reasons in response to a convincing rebuttal by the authors.

Suggestions for Potential Improvements:
1. Broader Benchmarking: Initially, reviewers criticized the use of only small datasets. Though later addressed, a more thorough inclusion of large and diverse benchmarks was a consistent request.
2. Comparison with Other Anti-Oversmoothing Methods: The paper could benefit from explicit comparisons with stochastic and residual methods in the anti-oversmoothing literature.
3. Graph Sparsification Context: The authors were advised to clarify how their method differs from or complements traditional graph sparsification techniques.
4. More Metrics: Reviewers suggested reporting pruning ratios and structural impact, which the authors included post-rebuttal. However, integrating this more prominently in the main paper was recommended.
5. Real-World & Domain-Specific Applications: Testing on chemical or physical datasets where oversmoothing is problematic was proposed as a direction for future work.
6. Clarity in Pseudocode & Practical Deployment Details: Some confusion over pseudocode was flagged; reviewers appreciated clarification but suggested cleaning up such artifacts in the final version.